# Macrophage and neutrophil heterogeneity at single-cell spatial resolution in human inflammatory bowel disease

Alba Garrido-Trigo[1,2], Ana M. Corraliza [1,2], Marisol Veny[1,2], Isabella Dotti[1,2], Elisa Melón-Ardanaz[1,2], Aina Rill [3], Helena L. Crowell [4], Ángel Corbí [5], Victoria Gudiño[1,2], Miriam Esteller[1,2], Iris Álvarez-Teubel[1,2], Daniel Aguilar[1,2], M. Carme Masamunt[1,2], Emily Killingbeck [6], Youngmi Kim[6], Michael Leon [6], Sudha Visvanathan[7], Domenica Marchese[8], Ginevra Caratù [8], Albert Martin-Cardona [2,9], Maria Esteve[2,9], Ingrid Ordás[1,2], Julian Panés[1,2], Elena Ricart[1,2], Elisabetta Mereu [3,11], Holger Heyn [8,10,11] & Azucena Salas[1,2] ✉

Ulcerative colitis and Crohn's disease are chronic inflammatory intestinal diseases with perplexing heterogeneity in disease manifestation and response to treatment. While the molecular basis for this heterogeneity remains uncharacterized, single-cell technologies allow us to explore the transcriptional states within tissues at an unprecedented resolution which could further understanding of these complex diseases. Here, we apply single-cell RNA-sequencing to human inflamed intestine and show that the largest differences among patients are present within the myeloid compartment including macrophages and neutrophils. Using spatial transcriptomics in human tissue at single-cell resolution (CosMx Spatial Molecular Imaging) we spatially localize each of the macrophage and neutrophil subsets identified by single-cell RNA-sequencing and unravel further macrophage diversity based on their tissue localization. Finally, single-cell RNA-sequencing combined with single-cell spatial analysis reveals a strong communication network involving macrophages and inflammatory fibroblasts. Our data sheds light on the cellular complexity of these diseases and points towards the myeloid and stromal compartments as important cellular subsets for understanding patient-to-patient heterogeneity.

Inflammatory bowel diseases (IBDs) are chronic immune-mediated diseases of the gastrointestinal tract that are normally classified as Crohn's disease (CD) or ulcerative colitis (UC) based on histological, imaging, and clinical features[1]. Despite this classification, a remarkable degree of variability is routinely observed in clinics in terms of disease severity, response to therapy and disease progression[2]. However, no validated clinical or biological features have been established to explain and faithfully predict such variability.

Single-cell RNA sequencing (scRNA-seq) of the intestinal mucosa recently provided a description of close to sixty different cell types present in UC[3–6] and ileal CD[7], emphasizing the magnitude of changes across cell populations in the context of intestinal inflammation. We applied scRNA-seq to colonic biopsies of healthy and active UC and colonic CD patients with a focus on understanding the heterogeneity among patients within each cellular compartment. The myeloid compartment, including macrophages and neutrophils, showed the

highest diversity in composition within patient groups, suggesting these cell types may contribute to differences among patients in disease phenotype and progression over time.

Macrophages are resident immune cells that act as gatekeepers in tissues and are well-known for their ability to sense and adapt to environmental changes. Two states of activation were initially described in mice, termed classical or M1 and alternative or M2 macrophages[8] that express different markers[9-11]. Importantly, signature genes for both subsets have been found in the human colon, including IBD samples[12], which showed a marked increase in M1 populations in those patients. Macrophage polarization has been more recently revisited showing a broader functional repertoire of this cell population, and important differences among intestinal macrophages phenotype and function have been linked to their spatial distribution within intestinal layers[13,14].

By combining scRNA-seq with the recently developed highly multiplexed CosMx Spatially Molecular Imaging (SMI) (NanoString Technologies) analysis[15], we discovered the signatures and tissue distribution of previously uncharacterized intestinal macrophage populations, including two subsets of resident and inflammation-related macrophages. This approach helped us understand the potential in vivo roles and likely interacting partners, including epithelial cells and fibroblasts, of the different macrophage subsets. Overall, our study emphasizes the diversity and plasticity of the intestinal myeloid compartment, specifically of the macrophage and neutrophil populations, and reveals mechanisms potentially contributing to heterogeneity in IBD.

## Results

### Integration of single-cell RNA sequencing and spatial molecular imaging analysis provides a map of healthy and inflamed colon

ScRNA-seq analysis of colonic biopsies from HC ($n = 6$), CD ($n = 6$) and UC ($n = 6$) active patients (Fig. 1a; Supplementary Table 1) totaling 46,700 cells identified 79 different clusters (Fig. 1b), whose proportions varied significantly between disease groups (Fig. 1b, c; Supplementary Fig. 1a). Each compartment (epithelium, stroma, B and plasma cells, T cells and myeloid cells) was isolated in silico to achieve higher resolution on cell populations. Analysis of differentially expressed genes (DEGs) detected cluster-specific marker genes with adjusted p-values. DEG for each cluster (gene list can be found at https://servidor2-ciberehd.upc.es/external/garrido/app/) were used to annotate subpopulations. Subsets, such as inflammatory fibroblasts, neutrophils, or inflammatory M1 macrophages, were found in some CD or UC patients, but not detected in HC (Supplementary Fig. 1a).

An additional cohort of formalin-fixed paraffin-embedded (FFPE) colonic samples (Fig. 1a; Supplementary Table 1, cohort 2) was processed using CosMx Spatial Molecular Imaging (SMI; NanoString Technologies)[15]. Scanner for Fields of View (FoVs) and immunofluorescence staining of pan-cell markers (CD45, PanCK, CD3) were performed on all tissues (Supplementary Fig. 2a, b). Selected FoVs were processed by CosMx SMI using a multiplex panel of 1000 genes. Annotation of cells was performed by label transfer based on scRNA-seq clusters, using the 100 top-ranked markers and count matrix, which assigned a unique label to each cell (Fig. 1d; Supplementary Fig. 1b and Supplementary Fig. 2). Markers for each cell type in the SMI dataset can be found at https://servidor2-ciberehd.upc.es/external/garrido/app/.

Using these datasets, we performed a correlation analysis of cell abundances based on scRNA-seq data (Supplementary Fig. 1c) and spatial correlation based on SMI data (Supplementary Fig. 1d). In IBD samples, we observed significant positive correlations in the abundances of cycling, germinal Center (GC) and memory B cells. Similarly, epithelial cell subsets tended to correlate in abundance with each other, while structural fibroblasts, including S1 (lamina propria), S2 (pericryptal), S3 (submucosal) and myofibroblasts (muscularis mucosa) were positively correlated (Supplementary Fig. 1c). Interestingly, we observed the significant correlation in the abundance of Glia and BEST4 OTOP2, Colonocytes and epithelium Ribhi, as well as M2 macrophages and S1 (lamina propria fibroblasts). While not significant, the abundance of glial cells was anti-correlated with most inflammatory cells including neutrophils, inflammatory fibroblasts, M1 macrophages and inflammatory monocytes. We propose that the presence of Glia is associated with a preserved intestinal architecture, while inflammation may lead to loss of this cell type. Regarding spatial co-localization of cell-types we observed marked architectural changes in inflamed tissues. In HC, cells in lymphoid structures (DCs, different types of T cells, FRCs, cycling cells, GC B cells, B cells and naïve B cells) were clearly spatially co-localized across samples (Neighborhood I). At least 3 other structural domains were found including a top lamina propria (Neighborhood II), a lower crypt (Neighborhood III) and a submucosa domain (Neighborhood IV) (Supplementary Fig. 1d). Remarkably, in inflamed samples, most neighborhoods were still represented but, in some cases, included inflammatory cells such as neutrophils (present in Neighborhoods I and II), or T cells CCL20 and Inflammation-Dependent Alternative (IDA) macrophages in neighborhood IV. A de novo neighborhood containing N3 neutrophils, M1 macrophages, inflammatory fibroblasts, and inflammatory monocytes (Neighborhood II) was found across IBD samples (Supplementary Fig. 1d).

Thus, by integrating scRNA-seq and CosMx SMI from human colonic samples, we have generated a map of healthy and diseased colon at single-cell spatial resolution.

### Transcriptional analysis at single-cell and spatial resolution reveals different populations of resident and inflammatory macrophages in the colonic mucosa

Remarkably, when comparing cluster proportions within patient groups, the largest discrepancies between individuals suffering from IBD were found within the myeloid, followed by the stromal compartment (Supplementary Figs. 1a, 3a), suggesting that the composition of these cell groups may be heavily influenced by patient-dependent factors, and could thus contribute to patient-to-patient heterogeneity. However, to find a potential association between a specific cell type, or combination of cell types, and any of the multiple clinical variables that can differ among patients was not possible at this time given the small sample size of our study.

Pooled HC, CD and UC scRNA-seq data identified different myeloid clusters including several macrophage subtypes, dendritic cells, inflammatory monocytes, mast cells, neutrophils, and eosinophils (markers found in Supplementary Fig. 3b and at https://servidor2-ciberehd.upc.es/external/garrido/app/), whose proportions changed in both UC and CD samples compared to controls (Fig. 2a). To perform a differential abundance test without relying on cell clustering we applied Milo, a tool which relies on k-nearest neighbor graphs[16], and we confirmed the differential abundance between groups in the myeloid compartment (Fig. 2b).

In healthy colon, resident macrophages (expressing C1QA, HLA-DRB1 and SELENOP, among other genes) were found in different transcriptional states (Supplementary Fig. 3b). These included macrophages expressing well-described M2-specific markers (i.e., CD163L1, CD209, FOLR2), annotated as M2 and M2.2, and hereafter referred to as M2. We annotated the other two clusters present in HC as M0 (M0 and M0_Rib[hi]), as they lacked M2 markers but highly expressed all other macrophage-specific genes (Fig. 2c and Supplementary Table 2), while showing low expression of PTPRC (CD45) (Supplementary Fig. 3c). Besides M0 and M2, samples from UC and CD contained inflammatory monocytes and activated macrophage clusters that we annotated as M1 ACOD1 and M1 CXCL5 (pooled as M1) and IDA macrophages (Fig. 2a, d).

All myeloid subsets were found in tissue sections analyzed by CosMx SMI, showing marked differences in abundance and spatial distribution depending on the patient and/or disease type (Fig. 2e). Distances of all myeloid subsets to the mucosal surface are shown in

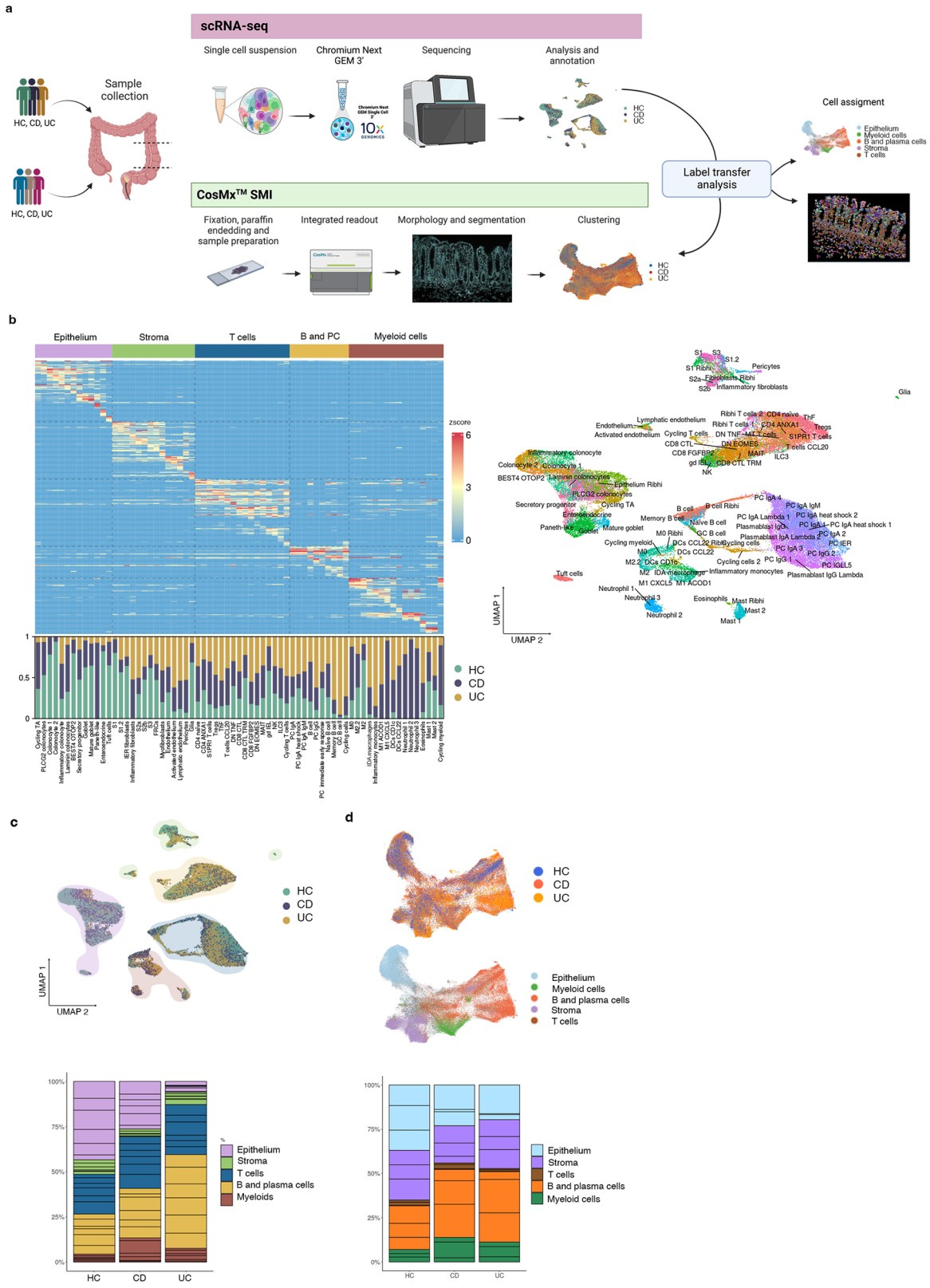

Supplementary Table 3. In healthy colon, resident macrophages were the closest to the mucosal surface whereas mast cells and DCs tended to localize in deeper mucosal layers. In inflamed samples, remodeling of the myeloid compartment was evident with M1 macrophages and neutrophils found closest to the mucosal surface (in ulcerated areas).

## M0 and M2 represent two independent states

Until now, M0 macrophages have not been formally described in the human intestine. Thus, we first compared our monocyte/macrophage signatures to publicly available data from HC, UC[3] and CD terminal ileum[7] (Supplementary Fig. 4a) and found, in both cohorts,

**Fig. 1 | Integration of single cell RNA sequencing (scRNA-seq) and Spatial Molecular Imaging (SMI) provides a map of healthy and inflammatory bowel disease (IBD) colonic biopsies. a** Overview of the study design for scRNA-seq, CosMx™ SMI and label transfer from scRNA-seq annotations to the SMI dataset. Two cohorts of colonic samples including active Crohn's disease (CD), active ulcerative colitis (UC) and healthy controls (HC) were processed by scRNA-seq (*n* = 18 samples) and CosMx™ SMI (*n* = 9 samples). Figure made in BioRender.com. **b** Heatmap of top marker genes discriminating the different cell subsets (epithelium, stroma, T cells, B and Plasma cells, and myeloid cells) and, below, barplots representing the proportions of each cell type resolved by scRNA-seq for HC, CD and UC. On the right, UMAP showing annotation of all cell types identified by

scRNA-seq. **c** UMAP and barplots of scRNA-seq data. Cells in UMAP are colored by group origin (HC, CD and UC) and clusters are shaded by cell subset (epithelium, stroma, T cells, B and Plasma cells, and myeloid cells). Barplots show the proportions of each cell subset in HC, CD and UC. Sub-segmentation of the barplots indicates the contribution of each individual. **d** UMAP and barplots of CosMx™ SMI data. Top UMAP shows cells colored by group (HC, CD and UC) while bottom UMAP is colored by cell subset (epithelium, stroma, T cells, B and Plasma cells and myeloid cells). Barplots show the proportions of each cell subset in HC, CD and UC. Sub-segmentation of the barplots indicates the contribution of each individual. Source data are provided as a Source Data file.

populations of macrophages that resembled the M0 and M2 subsets (Jaccard indexes = 0.3) (Supplementary Fig. 4b).

In agreement with the scRNA-seq data, we could visualize both CD209+CD68+ (M2) and CD209−CD68+ (M0) cells using immunostaining in healthy colonic lamina propria, mostly localizing below the apical epithelium (Fig. 2f) and also present throughout the lamina propria. Likewise, M0 and M2 cells were identified by CosMx SMI analysis (Fig. 2), confirming the dual identity of resident macrophages.

To understand their phylogenetic origin and their relation to other previously described macrophage subsets, we mapped our dataset to a recently published human monocyte-macrophage database containing data from 41 studies on several organs and diseases (MoMac-VERSE)[17] (Supplementary Fig. 4c). M0, M2.2 and M2 mapped to independent macrophage clusters within the MoMac-Verse dataset, supporting the hypothesis that they do represent two unique states.

Indeed, comparison of M0 and M2 macrophages in our dataset to in vitro monocyte-derived macrophages from published datasets[18] showed high similarity between intestinal M2 and in vitro M-CSF monocyte-derived macrophages (Supplementary Fig. 5a), while no or little overlapping with M0 macrophages was observed under these same conditions.

Interestingly, trajectory analysis of our data (Supplementary Fig. 5b) and the two public datasets annotated above (Supplementary Fig. 4d) suggests separate pseudo-time states for M0 and M2 clusters. In our data, M2.2 clusters, which express M2 markers, appear close to M0, suggesting they may represent a transitional state between the two resident compartments.

Overall, we conclude that in the healthy colon, resident macrophages are found in at least two states. M2 macrophages could potentially originate from circulating monocytes exposed to M-CSF in tissues, while the origin of M0 macrophages remains unknown.

**Inflammation-associated macrophages, including M1 and Inflammation-Dependent Alternative macrophages, show highly heterogeneous signatures among patients**

Compared to HC, IBD patients showed a marked increase in the total number and transcriptional heterogeneity of the macrophage population (Fig. 2a and Supplementary Fig. 1). Apart from M0 and M2, we found inflammatory/activated macrophages in at least three different states: two transcriptionally different M1 populations (M1 ACOD1 and M1 CXCL5) and the IDA macrophage cluster, in addition to a population of inflammatory monocytes. Comparison of these inflammation-associated cell types to the MoMac VERSE data set[17] is also shown in Supplementary Fig. 4d.

In contrast to M0 and M2 macrophages, the similarities between inflammatory macrophages in our cohort and those found in other intestinal datasets[3,7] were weaker, suggesting that activated macrophages may be found in highly patient/context-dependent states (Jaccard Index < = 0.14 Smillie et al, and Jaccard Index < = 0.16 Martin et al. Supplementary Fig. 4c).

As with M2 and M-CSF-derived macrophages, intestinal M1 CXCL5 cells showed high similarity to the in vitro GM-CSF-derived macrophages, while the signature of M1 ACOD1 was shared by both M-CSF

and GM-CSF-derived macrophages further stimulated with LPS[18] (Supplementary Fig. 5c). Remarkably, IDA macrophages showed the most transcriptional similarity to M-CSF-derived macrophages treated with serotonin (5-HT)[19] (Supplementary Fig. 5c).

Based on trajectory analysis, M1 subsets populated a different branch to those of M2/M0 subsets in all 3 datasets (Supplementary Figs. 4d, 5b), with inflammatory monocytes exclusively transitioning towards the fully activated M1 state. IDA macrophages instead appear to contain a heterogeneous population divided between the M1 and the M2 branches, suggesting they may represent a transitional state between those subsets. Analysis of overlapping markers between M1, M2 and IDA macrophages reveals that the latter share about 16% and 10% of its top 200 marker genes with M2 and M1, respectively (Supplementary Fig. 5d). However, analysis of DEG between IDA and the other two main macrophage populations revealed significant differences in their transcriptional profiles (Supplementary Fig. 5e and Supplementary Table 2).

Overall, we show that in the context of inflammation, macrophages can adopt diverse transcriptional signatures, with high heterogeneity between patients. Our data also suggests that intestinal macrophages could originate from monocytes activated under different stimuli including GM-CSF, GM-CSF + LPS, M-CSF + LPS or M-CSF + 5-HT, highlighting the importance of the microenvironment in modulating their phenotypes.

**Inflammation-dependent alternative macrophages express neuregulin 1**

Markers of IDA macrophages (gene list found at https://servidor2-ciberehd.upc.es/external/garrido/app/) include epidermal growth factor receptor (*EGFR*) family ligands like *AREG* and *HBEGF* and specifically *NRG1*, while showing lower expression of M1 and M2 canonical markers (Fig. 3a).

In agreement with scRNA-seq data (Fig. 3b), *NRG1* was significantly increased in bulk RNA-seq analysis of UC colonic mucosa compared to HC and CD (Fig. 3c), suggesting the potential involvement of neuregulin 1 in these patients. In situ hybridization of *NRG1* mRNA confirmed its expression on abundant CD68+ macrophages in IBD (Fig. 3d). In contrast, *NRG1* expression in HC was more specific to a population underlying the surface epithelium, with no or little colocalization within CD68+ macrophages (Fig. 3d). Indeed, our scRNA-seq and CosMx SMI analysis showed that in HC mucosa, S2b (*SOX6*+) (localized at the most apical area), but not S2a pericryptal fibroblasts (Supplementary Fig. 6d, e), also express *NRG1* (Supplementary Fig. 6a–c). In addition, fibroblast-expression of *NRG1* was also markedly increased in UC (Supplementary Fig. 6b, f).

Neuregulin 1 binds to ErbB receptors[20] and can promote epithelial differentiation to secretory lineages in human[21] and stem cell proliferation and regeneration in mice[22]. Using intestinal epithelial stem cell-enriched organoids, we found that neuregulin 1 significantly decreased expression of the stem cell marker *LRG5* and upregulated *OLFM4*, despite inducing no changes in the proliferation marker *MKI67* (Fig. 3e). Other signals, including NOTCH ligands and TNFα can induce *OLFM4* expression in epithelial cell lines[23]. We examined expression of NOTCH target genes and NOTCH receptors (*NOTCH1-4*) within

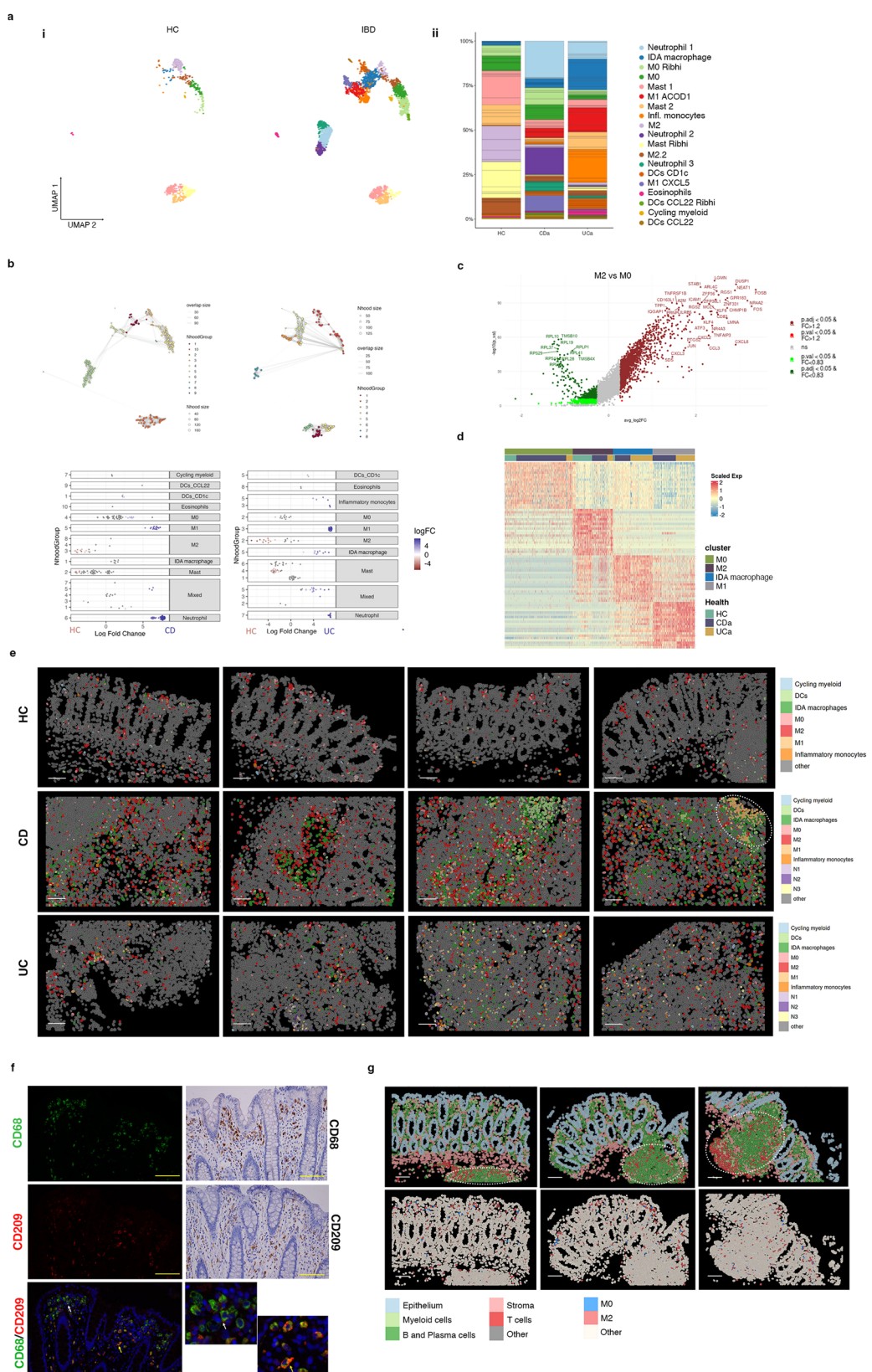

intestinal epithelium. *HES1* and *NOTCH1* were expressed in the intestinal epithelial compartment (Supplementary Fig. 7a). In contrast to *HES1*, *NOTCH1* expression was lower in fully differentiated epithelial cell types. Regarding TNF signaling, TNF receptor *TNFRSF1A* as well as the TNF target gene *TNFAIP3* were expressed by epithelial cells (Supplementary Fig. 7a), preferentially within differentiated colonocyte

populations, including inflammatory colonocytes, in addition to secretory progenitors (Supplementary Fig. 7a). To confirm these findings, we also investigated the spatial localization of *NOTCH1* and *TNFRSF1A* (found within the 1000-gene CosMx panel) which confirmed the different localization along the intestinal crypt of NOTCH and TNF receptors (Supplementary Fig. 7b). From these results alone we cannot

**Fig. 2 | Analysis of myeloid cell subsets in healthy and inflamed colonic mucosa. a** UMAP representation of scRNA-seq data for the myeloid clusters in healthy controls (HC, *n* = 6) and IBD colonic samples (CD *n* = 6, UC *n* = 6) (i); myeloid cell subset proportions across healthy and IBD samples (ii). **b** Cell type enrichment analysis using the differential abundance test Milo comparing CD (left panels) or UC (right panels) to HC. Nhood groups are shown for each comparison (top panels). The fold changes and annotations of each nhood in CD and UC samples are shown in lower panels. **c** Volcano plot of differentially expressed genes (DEGs) comparing M2 to M0 macrophages (see Supplementary Table 2 for the complete gene list). A two-sided Wilcoxon rank sum test was applied. Genes with a false discovery rate adjusted p value < 0.05, and a fold change (FC) > 1.2 or FC < 0.83 are considered regulated. **d** Heatmap showing average expression of DEGs for M0, M2, M1 and IDA macrophages in HC, CD and UC. **e** CosMx™ SMI images showing spatial distribution of the different myeloid cell populations in representative Fields of

View of colonic tissue of two HC, one CD and two UC patients. White dotted circle indicates ulcerated area with abundant M1 macrophages. Scale bar = 100 μm. **f** Double immunofluorescence showing M2 (CD209⁺CD68⁺) and M0 (CD209⁻CD68⁺) cells in one healthy tissue (left and lower right insets). White and yellow arrows indicate M0 and M2 macrophages, respectively. Right two top panels: immunohistochemistry showing CD68 and CD209 expression in one healthy tissue. Images are representative of 12 independent samples. Scale bar = 20 μm. **g** CosMx™ SMI images of a representative healthy colonic sample with lymphoid follicles (highlighted by dotted circles). The cellular localization of epithelial, stroma, T cells, B and plasma cells, and myeloid cells,(top panels) and M0 and M2 macrophages (lower panels) is shown. Scale bar = 100 μm. Source data are provided as a Source Data file. HC, healthy controls. IBD, inflammatory bowel disease. CD, Crohn's disease. UC, ulcerative colitis.

conclude a direct effect of these ligands on *OLFM4* expression. Nonetheless, we suggest a potential role of NOTCH signaling on expression of *OLFM4*, based on preferential expression of both *OLFM4* and *NOTCH1*, to a progenitor cell type in healthy tissues (Epithelium Ribhi, Secretory Progenitors) (Fig. 3f; Supplementary Fig. 8a–c).

In UC and CD, where *OLFM4* expression is dramatically increased, as shown here using scRNAseq (Fig. 3f), bulk RNA analysis (Fig. 3g), immunostaining, in situ hybridization, and CosMx™ SMI of colonic tissue (Fig. 3h), and by others[24], we propose instead that the upregulation of neuregulin 1 could contribute to this increase in *OLFM4* production.

Besides *OLFM4*, through SMI we observed other changes that occur in the intestinal epithelium of IBD patients, including the upregulation of anti-microbial mechanisms such as the expression of defensins (*DEFA5*), lipocalins (*LCN2*) and enzymes involved in producing reactive oxygen species (*DUOXA2*) (Fig. 3i; Supplementary Fig. 8d, e).

In summary, we show that IDA macrophages and S2b fibroblasts overexpress *NRG1* in IBD, particularly UC patients. Neuregulin 1, among other roles, promotes the expansion of the transit-amplifying epithelial compartment, which could play a role in the regeneration of the epithelium.

## CosMx Spatial Molecular Imaging analysis confirms the expansion of Inflammation-Dependent Alternative macrophages and reveals their tissue distribution in inflammatory bowel disease colon

CosMx SMI analysis localized abundant IDA macrophages scattered throughout the inflamed (UC and CD) colon, representing the most expanded inflammation-dependent macrophage state (Fig. 4a). In contrast, M1 macrophages were less abundant in the lamina propria and submucosa, but predominated within surface ulcers (Fig. 2g). Of note, in one CD patient we found abundant granulomas (Fig. 4b and Supplementary Fig. 9a). Granulomas are aggregates of macrophages, including multiploidy macrophages, which develop in response to persistent inflammation/infection and that are a pathological feature found in about one-fourth of CD patients[25]. IDA macrophages, together with some M2, and a few M0 and M1 macrophages, were the predominant macrophage state within granulomas (Fig. 4b), which were surrounded by diverse lymphoid subsets (Supplementary Fig. 9b). The cellular composition of a non-granuloma lymphoid aggregate in the same patient is shown for comparison (Supplementary Fig. 9c).

In agreement with the CosMx SMI results, immunostaining showed low and scattered staining of the M2 (CD209) markers within CD68⁺ cells in granuloma (Fig. 4b, c and Supplementary Fig. 9a). Compared to lamina propria macrophages, *NRG1* expression within the granulomas was low (Fig. 4c).

Thus, IDA macrophages abundantly present in the inflamed colon display differential *NRG1* expression depending on their tissue location. While *NRG1*ʰⁱ IDA macrophages localize to the most apical subepithelial compartment of the mucosa, *NRG1*ˡᵒʷ alternatively activated

macrophages accumulate within granulomas in CD and in the submucosa of both UC and CD patients, suggesting independent functions.

## Inflammatory fibroblasts co-localize with Inflammation-Dependent Alternative macrophages in inflammatory bowel disease

Given the abundant number and heterogeneity in distribution patterns of IDA macrophages in UC and CD patients, we leveraged the multiplexed spatial data to identify the cell types that were most frequently found in their proximity. IDA macrophages tended to localize near to other macrophage subsets (M0, M2 and M1), some stromal cells, and T cells, particularly CD8⁺ T cells, Tregs and T cells CCL20 (Fig. 5a). Within the stromal compartment, IDA macrophages presented high spatial correlation with inflammatory fibroblasts in both UC and CD (Fig. 5b, c), including within granulomas (Supplementary Fig. 10). Inflammatory fibroblasts were described in UC[5] and found here in colonic CD (Fig. 5d), as confirmed by biopsy bulk RNAseq (Fig. 5e). Importantly, inflammatory fibroblasts expressed *CSF2* and *CSF3*, encoding for GM-CSF and G-CSF, respectively, and prostaglandin-producing enzymes *PTGS1, PTGES, PTGS2* (Fig. 5f). In fact, a recent study[26] showed that prostaglandins are produced by activated fibroblasts and drive the differentiation of IDA-like macrophages expressing *HBEGF* and *EREG*, but not *NRG1*, in the synovium of rheumatoid arthritis patients.

Thus, we argue that a crosstalk between inflammatory fibroblasts and macrophages may take place in IBD via specific ligand-receptor interactions.

## Single-cell RNA sequencing reveals a marked heterogeneity of tissue neutrophils in inflammatory bowel disease colonic mucosa

Finally, in addition to the heterogeneity within macrophages in IBD, we also found diverse populations of intestinal granulocytes using scRNA-seq (Fig. 2a). Granulocytes, including eosinophils and neutrophils, increased in IBD (Supplementary Fig. 11a–c) and expressed distinct membrane protein markers (*CD62L, CD193, CD69*) compared to their peripheral counterparts, indicating different states of activation (Supplementary Fig. 11d). Specific eosinophil markers included *CLC, MS4A3, CCR3* and the "Th2" cytokines IL4 and IL13 (Supplementary Fig. 3a), while the *HCAR3, FCGR3B* (CD16b), *CMTM2* and *PROK2* were specific neutrophil markers (Fig. 2b). Supporting their increase in UC and CD, the expression of most eosinophil and neutrophil markers was significantly increased in bulk biopsy RNAseq (Supplementary Fig. 11e).

Intestinal neutrophils were found in 3 unique states (annotated as N1, N2 and N3) whose relative abundance varied on individual patients and disease type (Fig. 6a). Compared to N1 and N3, N2 neutrophils, instead expressed higher levels of *CCL3, LGALS3* and *CXCR4* (Fig. 6b, c), while N3 neutrophils displayed a marked IFN-response signature (e.g. *GBP1, IRF1* and *FCGR1A*).

Compared to public scRNA-seq datasets from peripheral neutrophils, N1 and N3 neutrophils showed the highest similarity to both

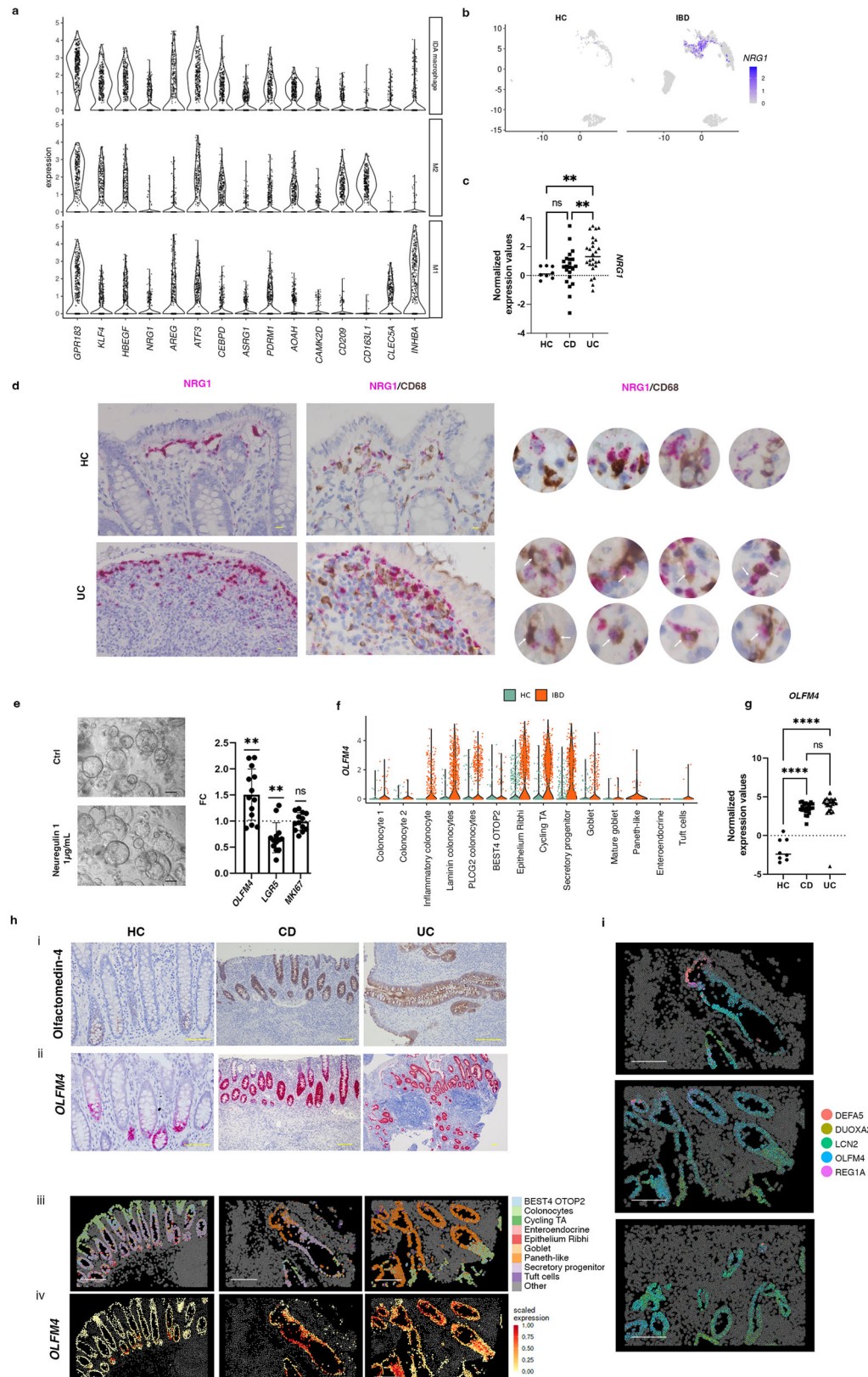

bone marrow mature neutrophils (BM matureN)[27] and to umbilical cord blood neutrophils (UCB)[28] (Fig. 6c). In contrast, N2 neutrophils showed little overlap with peripheral neutrophils and instead expressed genes suggestive of different tissular locations (e.g., chemokines and receptors, as mentioned above) and different activation states/functions (*CD83, CD63, FTH1, VEGFA*). Protein expression of two of

these N2 markers (CXCR4 and CD63) was also confirmed for 10% and 61% of tissue neutrophils compared to 0.26% and 14% of blood neutrophils, respectively (Supplementary Fig. 11d).

All 3 neutrophil subsets were found using CosMx SMI and showed scattered distribution throughout inflamed lamina propria, with predominant localization in crypt abscesses and ulcerated areas (Fig. 6d).

**Fig. 3 | Neuregulin 1 expression and function in colonic mucosa. a** Violin plot showing the expression of selected marker genes on IDA, M2 and M1 macrophages from pooled HC ($n = 6$), CD ($n = 6$) and UC ($n = 6$) scRNA-seq data. **b** UMAP showing *NRG1* expression in the myeloid compartment of HC and IBD data. **c** *NRG1* expression from bulk biopsy RNA-seq data in HC ($n = 8$), and active CD ($n = 22$) and UC ($n = 26$) patients. Ordinary one-way ANOVA test (Benjamini-Yekutieli) was performed correcting for multiple comparisons. Each sample is represented as a dot and the median value as a line. **$p < 0,0068$, ns = 0.1693. **d** Double In situ hybridization of *NRG1* and immunohistochemistry for CD68 in one HC and one UC sample (representative image out of 6 independent biological replicates). White arrows show NRG1+CD68+ cells. Scale bar = 10 μm. **e** Human-derived epithelial organoids treated with vehicle (Ctrl) or Neuregulin 1 (1 μg/mL) for 48 h. Scale bars = 100 μm. mRNA expression of *OLFM4*, *LGR5* and *MKI67* was determined by RT-qPCR ($n = 13$ biologically independent samples). One-sample t-test was performed. Data is shown as fold change (FC) relative to the vehicle-treated condition. Bars represent mean ± standard deviation. *OLFM4* $p = 0,0029$ (**), *LGR5* $p = 0,0019$(**), *MKI67* ns:

$p = 0,8518$. **f** Violin plot showing *OLFM4* expression by scRNA-seq in all epithelial cell subsets from HC (green) and IBD (orange) samples. **g** *OLFM4* expression by bulk RNA-seq of colonic samples from HC ($n = 8$), active CD ($n = 22$) and active UC ($n = 26$) patients. Ordinary one-way ANOVA test (Benjamini-Yekutieli) was performed correcting for multiple comparisons. Each sample is represented as a dot and the median value as a line. $p < 0,0001$(****), ns: not significant. **h**, (i) Olfactomedin 4 immunostaining and (ii) *OLFM4* in situ hybridization in HC, CD and UC colon (images representative of 9 biological replicates). Scale bars = 100 μm. (iii) CosMx™ SMI visualization of epithelial cell subsets in a HC and two UC representative Fields of View (FoVs) and (iv) mean expression of *OLFM4* in each of those cells analyzed by CosMx™ SMI. Scale bar = 200 μm. **i** Expression of *DEFA5*, *LCN2*, *DUOX2A*, *REG1A*, and *OLFM4* within epithelium of representative FoVs of a UC patient analyzed by CosMx™ SMI. Dots represent mRNA molecules. Scale bar = 200 μm. Source data are provided as a Source Data file. HC, healthy controls. IBD, inflammatory bowel disease. CD, Crohn's disease. UC, ulcerative colitis.

## Discussion

ScRNA-seq has boosted the resolution at which complex tissues, including the inflamed intestine, can be studied[3-7]. Nonetheless, available scRNA-seq datasets lack information on tissue distribution and spatially relevant cell-to-cell interactions. To fill this critical gap, highly multiplexed spatial technologies are rapidly evolving[29]. Our study provides a dataset that combines scRNA-seq data with spatial transcriptomics at single-cell resolution to start unraveling patient-dependent disease mechanisms.

We focused on the myeloid compartment, including both macrophage and neutrophil subsets, as they showed the highest degree of variation within patient groups. We argued that changes in these populations may explain disease heterogeneity. Macrophages are well-known for their tissue plasticity. The origin, phenotype, and function of intestinal macrophages, however, continues to be a subject of debate[30,31]. Nonetheless, resident macrophages are known to display heterogenous functions[13,32]. In the context of IBD, activated subsets have been described as having a proinflammatory function[12,33]. Cell classification, however, relies on surface markers that may not be consistently used across studies, thus making standardization challenging. ScRNA-seq provides instead unbiased whole transcriptome profiles of cell types, independently of prior knowledge of marker expression. Using unsorted cells, we discovered at least two unique resident macrophage states (M0 and M2) in healthy colonic mucosa. Both subsets were still present in active patients, together with a variety of activated inflammatory macrophages. Remarkably, the profiles of M0 and M2 macrophages were consistently found in two independent datasets, including an ileal CD cohort[3,7] and localized by CosMx SMI to the intestinal lamina propria. In contrast, the transcriptional signatures of inflammation-associated macrophages varied markedly between patients and datasets. We argue that, compared to canonically differentiated resident macrophages, inflammatory macrophages adapt their phenotype to a variety of patient-dependent microenvironments. Based on published data from in vitro differentiated macrophages and pseudo-time analysis of scRNA-seq signatures, we also propose that both infiltrating monocytes (found in inflamed samples) and resident M2 macrophages may give rise to activated macrophages. Multiplexed spatial analysis confirmed the diversity in the macrophage populations, and importantly showed that most inflammation-dependent macrophages do not display the full characteristic M1-signature exhibiting instead an alternative activation pattern characterized by the expression of EGFR ligands, *NRG1* and *HBEGF*, and the C-type lectin receptors *CLEC10A* and *ASGR1*. While an M1 signature can be reproduced in vitro by exposure of blood monocytes to GM-CSF, GM-CSF/LPS or M-CSF/LPS, the origin of IDA macrophages remains incompletely understood. We found that an endogenously produced factor, serotonin, which is highly abundant in the gut, can induce in vitro a signature on M-CSF-derived

macrophages (M2-like) that resembles that of the IDA subset found in IBD. While previous studies have shown that serotonin, primarily produced by enterochromaffin cells, modulates macrophage cytokine signatures[34], our data does not prove this to be a relevant mechanism in patients, nor does it rule out the existence of other signals, including fibroblast-derived prostaglandins[26], that could drive this alternative activation. To our knowledge, macrophages expressing neuregulin 1 have only been described in a murine model of myocardial infarction[35] and suggested to prevent the progression of fibrosis in mouse hearts. The functional role of IDA macrophages in our patients, thus, remains unclear. *NRG1*hi IDA macrophages tended to localize to the most apical side of the mucosa and could potentially play a role in epithelial regeneration based on their ability to produce EGFR ligands, which can act on the intestinal epithelium and drive transition towards a regenerative (*OLFM4*-expressing) phenotype. In contrast, *NRG1*low IDA macrophages were found within granulomas of CD patients and in the submucosa of inflamed patients, suggesting IDA macrophages may play different roles depending on their environment.

Based on both scRNA-seq and SMI, we hypothesize that the interaction between M2 or IDA macrophages and inflammatory fibroblasts could play a role in disease pathophysiology. While there is abundant data in the literature to support the interaction of macrophages and fibroblasts, particularly in the context of cancer and fibrosis[36], little is known about their crosstalk in the context of chronic inflammation. Inflammatory fibroblasts represent a disease-specific fibroblast subset characterized by the expression of multiple cytokines including profibrotic *IL11*, *IL24*, *IL8*, *IL6*, *TGFβ1*, and tissue remodeling metallopeptidases, making them attractive therapeutic targets to manage inflammation and potentially, fibrosis, a common and difficult-to-treat complication of chronic intestinal inflammation. Emphasizing their interaction with macrophages, inflammatory fibroblasts express *CSF2* (GM-CSF), which promotes macrophage activation, while activated macrophages can produce mediators (i.e., *OSM*, *IL6*, *TNF*, etc.) that can drive fibroblast activation. Furthermore, besides its role on epithelial regeneration, EGFR signaling is a robust regulator of fibroblast motility[37] and may be involved in cartilage and bone destruction in rheumatoid arthritis[26].

Beyond macrophages, recent scRNA-seq studies have explored the diversity and plasticity of blood neutrophils[38,39]. These short-lived cells, originally thought to exist in fixed states, have more recently been shown to be transcriptionally dynamic, adopting multiple transcriptional states depending on their maturation stage. Compared to available data on periphery, intestinal neutrophils, including a subset that shows a signature of IFN-inducible genes, express the maturation marker CXCR2[38]. Remarkably, CXCR4 neutrophils (N2) were not found in peripheral datasets. CXCR4, which is essential for bone marrow retention of immature neutrophils, has been identified in mice to mark

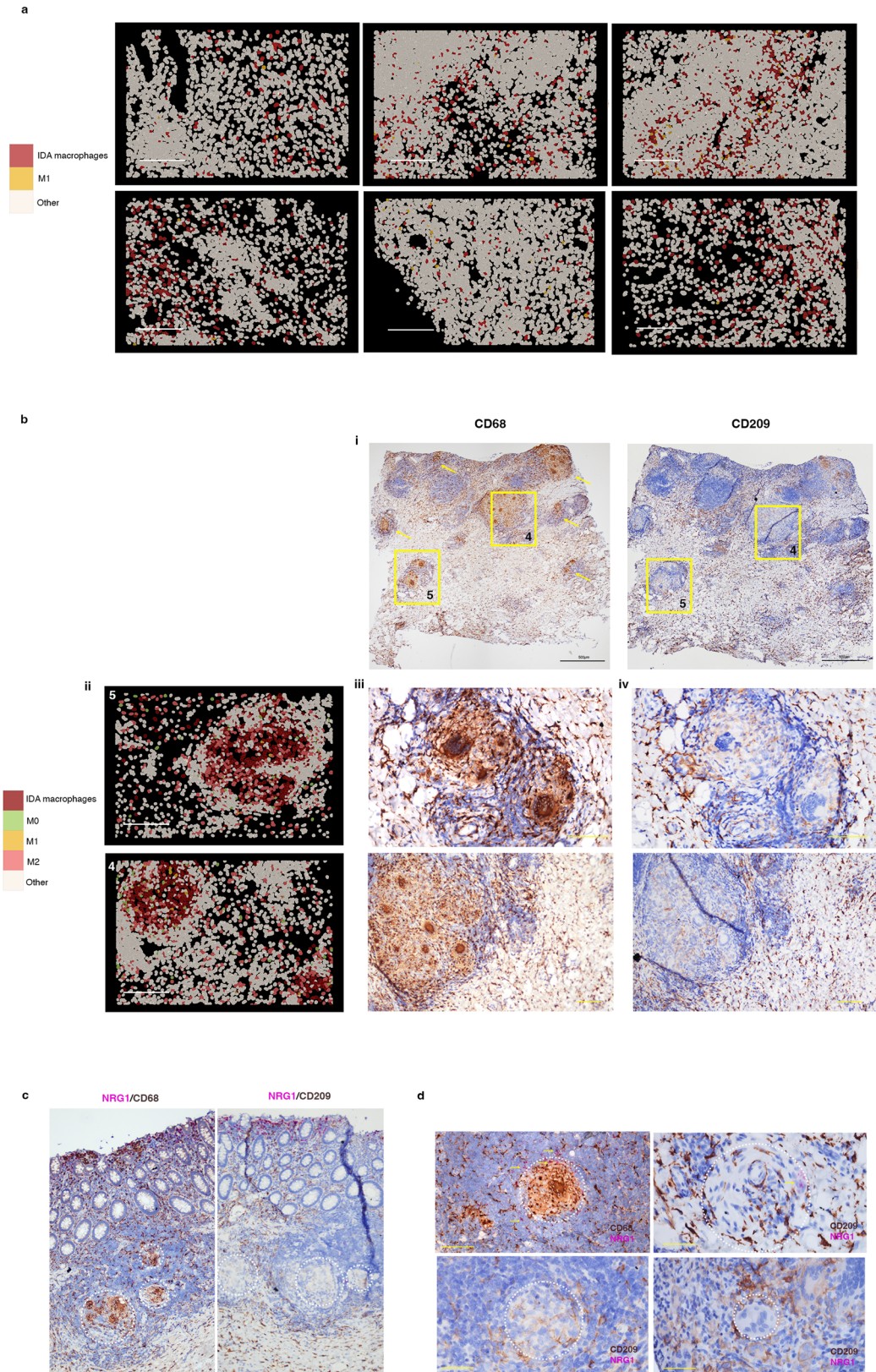

a subset of pro-angiogenic neutrophils found both in lung and intestine[40], and to be expressed by neutrophils in inflamed tissues[41,42]. While evidence of angiogenic neutrophils in humans remains elusive, our data points towards the presence of this neutrophil subset, at least in inflamed tissues. Further analysis is required to fully understand the origin and function of these neutrophils in the gut.

Despite the important information that can be drawn from our datasets, there are a few limitations to our study that must be considered. First, while we show high diversity in cell composition within the myeloid and stromal compartments among different patients, we cannot provide a direct link between the different cellular composition and specific disease features (i.e. disease phenotype, responsiveness to

**Fig. 4 | Inflammation-Dependent Alternative (IDA) macrophages are widely distributed in ulcerative colitis and present in Crohn's disease granulomas.**
**a** CosMx™ SMI distribution of IDA and M1 macrophages in IBD colonic samples. Images from 1 representative UC (2 Fields of View (FoV)) and 1 CD (4 FoVs) patients (out of a total of 6 patients analyzed) are shown. Scale bar = 200 μm. **b** CD colonic sample with multiple granulomas (CD b patient). (i) Two FoVs from this patient are indicated by squares and other granulomas found in the same sample by yellow arrows. (ii) Macrophages within granulomas are shown by CosMx™ SMI in FoVs 4 and 5 and protein expression (Scale bar = 200 μm) of (iii) CD68 and (iv) CD209 is shown by immunohistochemistry on the same tissue sections (Scale bar = 100 μm).

**c** Double *NRG1* in situ hybridization and CD68 or CD209 immunostaining in tissue sections from the CD patient (CD b) containing abundant granulomas. In situ hybridization of *NRG1* shows an increasing gradient of expression towards the apical mucosa. Granulomas are indicated by dotted circles (Scale bar = 100 μm). **d** Magnified pictures of representative granulomas of the same CD tissue stained for *NRG1* using in situ hybridization and CD68 or CD209 immunostaining. Granulomas are indicated by dotted circles and *NRG1* positive cells are shown by arrows (Scale bar = 100 μm). Source data are provided as a Source Data file. HC, healthy controls. IBD, inflammatory bowel disease. CD, Crohn's disease. UC, ulcerative colitis.

---

drugs or long-term prognosis) due to the small sample size of our study. Hence assessing the clinical relevance of our findings will require a larger cohort or access to additional data sets with complete associated metadata which is currently unavailable.

In addition, our study is limited to colonic disease and does not consider the peculiarities of ileal disease, common in CD patients. Nonetheless, comparison with a published ileal CD cohort[7] showed remarkable similarities in macrophage heterogeneity compared to our colonic dataset. In addition, the 1000-gene SMI panel used in our study, while sufficiently large to cover most cell types, lacked important markers that may have limited our accuracy when assigning cell identities. This may be especially true for cell subsets sharing most of their transcriptomic signature (i.e., N1, N2 and N3 neutrophils).

Overall, we provide evidence to support high patient-dependent heterogeneity within the myeloid compartment in both UC and colonic CD. We argue that intestinal macrophages, which sense changes in the microenvironment, could act as reliable indicators of patient-specific molecular patterns and thus, promising targets. Furthermore, we show that by combining scRNA-seq with SMI, cell subsets can be assigned to likely interacting partners, thus providing crucial niche information. This spatial resolution will be essential in understanding cellular function, and to faithfully link biologically relevant interactions to specific cell types.

## Methods

### Patient recruitment and sample collection
Colon samples were collected to perform scRNAseq, generate organoid cultures and analyze by flow cytometry on fresh tissue. Additional samples were fixed in formalin and paraffin embedded (FFPE) for CosMx™ SMI and tissue staining (IHC, IF, ISH). For scRNAseq and flow cytometry, colon biopsies from active areas of UC and CD patients were collected during routine endoscopies performed as standard of care. Healthy controls were individuals undergoing endoscopy for colorectal cancer screening as standard of care and presenting no signs of dysplasia or polyps at the time of endoscopy. For SMI analysis, surgical colon resections were obtained from non-IBD controls (patients undergoing surgery for colorectal cancer), UC and CD patients (undergoing colonic resective surgery). Organoid cultures were established exclusively from surgical samples of non-IBD controls. In non-IBD controls samples were obtained from non-tumor areas and for UC and CD patients from involved inflamed areas. Blood samples from HC, CD and UC were also collected to perform flow cytometry analysis of granulocytes. The study was approved by the Ethics Committee of Hospital Clinic Barcelona (HCB/2018/1062 and HCB/2022/0125) and the Hospital Mutua de Terrassa (CI201901). All patients signed an informed consent at the time of colonoscopy or before surgical intervention. Participants received no compensation for the study.

### Human colonic cell isolation for scRNAseq or flow cytometry analysis
Biopsies (n = 4–6 per patient) were taken from involved areas of the colon of UC (n = 6 for scRNA-seq and n = 12 for flow cytometry) and CD (n = 6 for scRNA-seq and n = 6 for flow cytometry) patients with signs of

endoscopic activity, placed immediately in cold Hank's Balanced Salt Solution (HBSS) (Gibco, MA, USA) and kept at 4 °C until processing (< 1 h). Colonic biopsies from non-IBD controls (n = 6 for scRNA-seq and n = 6 for flow cytometry) were collected from the sigmoid colon and processed in the same way. Freshly collected biopsies were washed with 5 mM DTT (Roche, Spain) in HBSS for 15 min and then washed in complete medium (CM) (RPMI 1640 medium (Lonza, MD, USA) supplemented with 10% heat-inactivated fetal bovine serum (FBS) (Biosera, France), 100 U/ml penicillin, 100 U/ml streptomycin and 250 ng/ml amphotericin B (Lonza), 10 μg/ml gentamicin sulfate (Lonza) and 1,5 mM Hepes (Lonza)) for 10 min. Both incubations were performed at room temperature in a platform rocker. Biopsies were chopped with a scalpel and placed into tubes containing 500 μl of Digestion Solution (CM + Liberase TM (0.5 Wünsch units/ml) (Roche, Spain) + DNase I (10 μg/mL) (Roche, Spain)) and incubated on a shaking platform for 1 h at 250 RPM and 37 °C. After incubation biopsies were filtered through a 50 μm cell strainer (CellTrics, Sysmex, USA), washed with Dulbecco's Phosphate Buffered Saline (PBS; Gibco, USA) and resuspended in RPMI medium supplemented with 0.05% of Bovine serum albumin (BSA) at a concentration of ~0.5–1·10⁶ cells/mL for scRNA-seq and in FACS buffer (PBS + 2% inactivated FBS (fetal bovine serum) + NaN3 0.1%) for flow cytometry analysis[43]

### 10× library preparation and sequencing
Following digestion, 10× Genomics 3' mRNA single-cell method was used. Approximately 7000 cells were loaded onto the Chromium10x Genomics platform (10x Genomics, CA, USA) to capture single cells, following the manufacturer's protocol. Generation of gel beads in emulsion (GEMs) (10x Genomics, CA, EEU), barcoding and GEM-reverse transcription was performed using the Chromium Single Cell 3' and Chromium Single Cell V(D)J Reagent Kits (10× Genomics, CA, EEU) (user guide, no. CG000086) according to manufacturer's instructions. Full-length, barcoded cDNA was amplified by PCR to generate enough mass for library construction (Nextera® PCR primers) (Illumina, CA, USA). Sequencing of the libraries was performed on HiSeq2500 (Illumina, CA, USA).

### Single cell data analysis
**Data processing.** For each sample, sequences obtained in fastq files were processed with CellRanger's count pipeline using the default parameters (10XGenomics, version 3.1.0). This pipeline performs an alignment based on the reference genome (Gencode release 27, assembly GRCh38 p10), filtering, barcode counting, and UMI counting. The resulting filtered matrix was analyzed using R (version 4.2.0). We merged the count matrices retrieved from CellRanger using the function merge from the SeuratObject R package (version 4.0.2). At this point, doublets were assessed using the scDblFinder[44] R package (version 1.8.0) and removed. Total analyzed cells: 47,600.

Initially, we analyzed healthy control and IBD samples separately to assess the similarity of cell types and samples. We processed and annotated the objects separately and we assessed for similarity by using Jaccard index and label transferring (see detailed methods in at https://servidor2-ciberehd.upc.es/external/garrido/methods1/). Samples were then pooled together in the same object. Low-quality cells

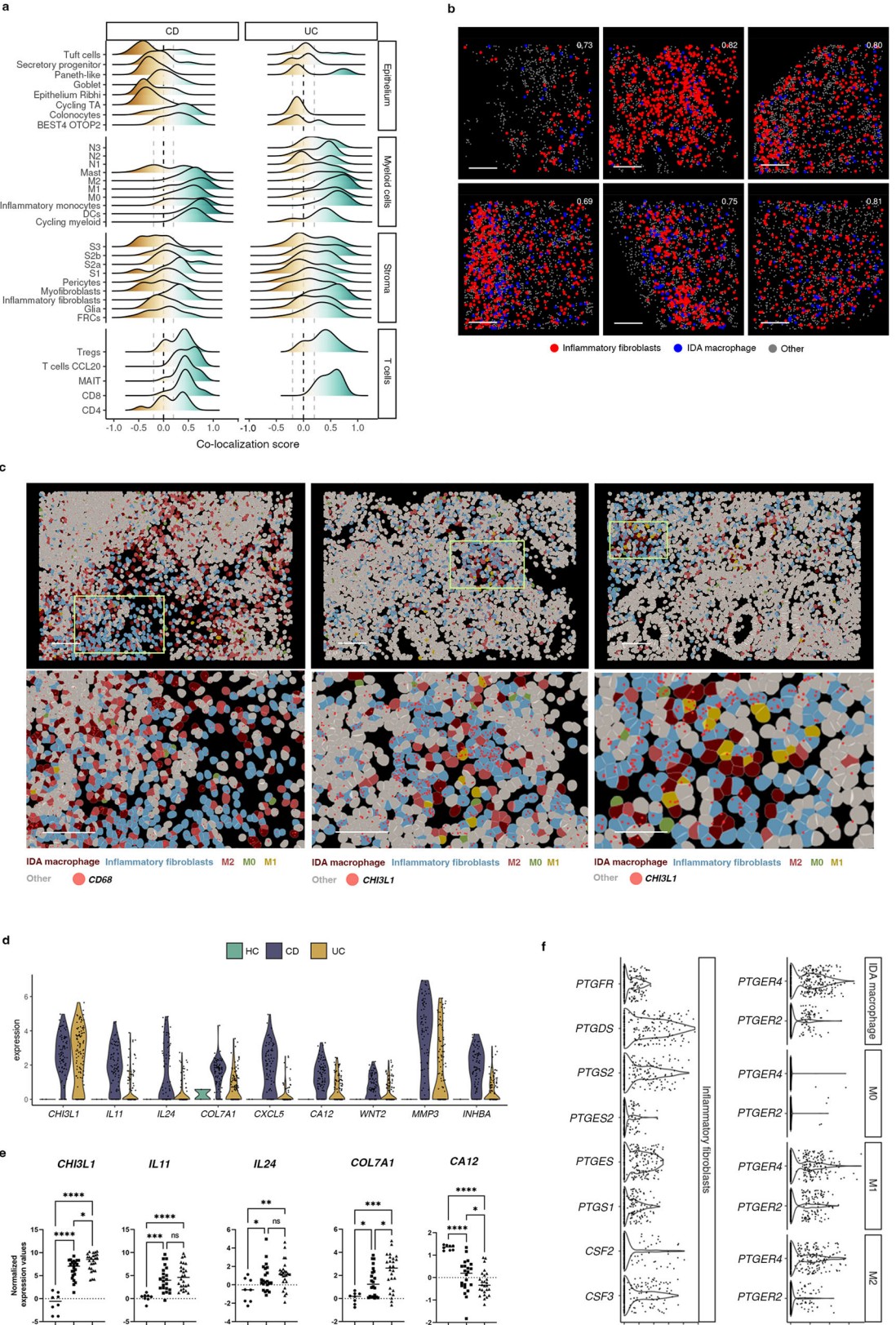

were then filtered out based on mitochondrial RNA percentage and number of genes per cell. Epithelial cells required a less stringent filter of 65% of counts aligned to the mitochondrial genes for quality control. A total of 46,700 cells were considered for the analysis. Then, we logarithmically normalized, obtained the highly variable genes, and scaled the counts (default parameters) of each data set using Seurat[45]

(version 4.1.0). Principal component analysis (PCA) was performed. Dimensionality reduction was performed by applying the Uniform Manifold Approximation and Projection (UMAP) algorithm using the optimal number of PCs[46]. UMAP also served as a two-dimensional embedding for data visualization. Cluster analysis was performed using the Louvain clustering algorithm[45].

**Fig. 5 | Inflammation-Dependent Associated (IDA) macrophages co-localize with inflammatory fibroblasts. a** Ridge plot of co-localization analysis of IDA macrophages and epithelial, other myeloid, stromal and T lymphocytes by CosMx™ SMI. Correlation for cell positions was calculated per cell type (0 indicates no correlation, >1 indicates co-localization with 1 being cells sharing the same position; <1 indicates negative correlation between the indicated cell types). Data is pooled from all CD (n = 3 donors, total of 56 Fields of View (FoVs)) and UC (n = 3 donors, 67 FoVs) patients. **b** Co-localization analysis between IDA macrophages and inflammatory fibroblasts in inflamed UC tissue. Six representative FoVs from 2 independent UC patients are shown. Co-localization scores are indicated in white for each FoV. Scale bar = 100 μm. **c** Images containing IDA macrophages and inflammatory fibroblasts. Three representative FoVs (from 2 patients, 1 UC and 1 CD) are shown. Expression of *CD68* (macrophages) or *CHI3L1* (inflammatory fibroblasts) is shown as red dots. Each dot represents a single mRNA molecule.

Scale bar = 100 μm. **d** Violin plots showing expression (y-axis) of marker genes (x-axis) of inflammatory fibroblasts in HC (n = 6), active CD (n = 6) and active UC (n = 6) determined by scRNA-seq. **e** Expression of markers of inflammatory fibroblasts in HC (n = 8), and active CD (n = 22) and UC (n = 26) patients in bulk biopsy RNA-seq data. Ordinary one-way ANOVA test (Benjamini-Yekutieli) was performed correcting for multiple comparisons. Each sample is represented as a dot and the median value as a line $p < 0.05$ (*), $p < 0.01$ (**), $p < 0.001$ (***), $p < 0.0001$ (****), ns: not significant. **f** Violin plot visualizing scRNAseq-based expression (y-axis) of prostaglandin-related genes in inflammatory fibroblasts, IDA macrophage, M2 (M2 & M2.2) and M1 (M1 ACOD1 & M1 CXCL5) in pooled data of HC (n = 6), CD (n = 6) and UC (n = 6). Expression of *CSF3* and *CSF2* in inflammatory fibroblasts has been also included. Source data are provided as a Source Data file. HC, healthy controls. IBD, inflammatory bowel disease. CD, Crohn's disease. UC, ulcerative colitis.

The resulting unsupervised 13 clusters were manually categorized into five main cell types: Epithelial cells, Lymphoid cells (T cells and innate lymphoid cells), Myeloid cells (mast cells, macrophages, dendritic cells, monocytes, neutrophils, and eosinophils), Stromal cells (endothelial cells, fibroblasts, pericytes and glia), and B and plasma cells. Markers used to identify each cell type are provided in Supplementary Table 4.

Immunoglobulin (IG) genes were removed from all the main cell types except B and plasma cells to reduce background noise.

**Identification of cell types.** Each main cell type was re-processed starting from FindVariableFeatures using the same procedure as for the whole dataset. During the process, doublets were identified through expert annotation of the marker gene lists for each cell cluster and corresponded to clusters with markers from distinct lineages. In brief, we removed *CD3, C1QA, DERL3,* or *MS4A1* expressing cells from epithelial cells; *CD3, C1QA,* or EPCAM-expressing cells from the B cells; *CD3, C1QA, DERL3, MS4A1* or *EPCAM*-expressing cells from the stromal cells; *DERL3, MS4A1, C1QA, EPCAM,* or *CD79A*-expressing cells from the lymphoid cells; and *CD3, THY1, DERL3* or *MS4A1*-expressing cells from the myeloid cells. We then systematically re-clustered each cell category using an unsupervised Louvain clustering algorithm. The annotation of each subcluster from the main cell type was defined by the marker genes, which were obtained by the FindAllMarkers function with the default threshold except for the min.pct parameter, which was set as 0.25, and the thresh.use, which was set to 0.25.

**Batch correction and cluster annotation.** To ensure the validity of our clustering results and identify any potential technical artifacts, we calculated the entropy for each cluster using the formula $H = 1 - \sum(p_i^2)$ where $H$ is the entropy, $p_i$ is the proportion of cells in cluster that belong to patient $i$, and the summation is taken over all samples in our experiment. Our comprehensive analysis, coupled with per-sample UMAP visualization, revealed the presence of batch effects by sample of lymphoid (tcells), B and plasma cells (plasmas), and stromal cells (stroma) during re-processing for cell type identification. To address these batch effects, we applied Harmony (version 0.1.1)[47] (version 0.1.0). Specifically, we used the RunHarmony function, with the optimal number of principal components found for each subset as latent space and the sample of origin as batch label. Seurat's analysis pipeline was applied again to each object, followed by UMAP generation using default settings and harmony integrated space. Cluster annotation was performed based on differentially expressed genes (Supplementary Table 5).

**Differential abundance testing.** We tested for differences in cell abundances between myeloid colonic cells from IBD patients and healthy controls using the miloR package (version 1.4.0)[16]. Specifically, we performed differential abundance testing of healthy patients and

both inflammatory chronic diseases, CD and UC, separately. To do so, the dataset was subsetted to include myeloid cells only (677 healthy cells, 1935 CD cells and 1161 UC myeloid cells). For each analysis, a k-nearest neighbors (KNN) graph was constructed using the graph slot from the adjacency matrix of the previously processed Seurat object and cells were assigned to neighborhoods ($k = 20, d = 30$). To leverage the variation in the number of cells between UC, CD and healthy samples, the cells belonging to each sample in each neighborhood were counted and the Euclidean distances between single cells in a neighborhood were calculated. Differential abundance testing in a generalized linear model framework was then performed between IBD samples and healthy controls with default parameters. In addition, to check if the differences in abundances are particularly strong for a certain cell type, we assigned to each neighborhood a specific annotation label considering most cells belonging to that neighborhood. For those neighborhoods where less than 80% of cells shared the most abundant label, a "mixed" label was assigned. Neighborhoods were grouped using the groupNhoods function with *max.lfc.delta = 5, overlap = 3* for CD and healthy samples, and *max.lfc.delta = 3, overlap = 2* for UC and healthy samples.

**Trajectory analysis.** We applied the Monocle 2 algorithm (version 2.24.1)[48] to order myeloid cells in pseudotime to indicate their developmental trajectories. Such pseudotime analysis is a measure of progress through biological processes based on their transcriptional similarities. With that aim, we selected all distinct macrophages and monocytes and created a *CellDataSet* object using a negative binomial model. We ran Monocle 2 using the 2000 most highly variable genes selected with Seurat (version 4.1.0) and default parameters of Monocle after DDRTree[49] dimension reduction and cell ordering. To visualize the ordered cells in the trajectory, we used the plot_cell_trajectory function to plot the minimum spanning tree on the cells. The starting point of the trajectory was set manually to M0 macrophages.

**Publicly available scRNAseq datasests.** Data from Smillie et al.[3] and Martin et al.[7] were downloaded and reanalyzed following the same protocol used for our samples. We separated in silico the macrophage (*CD14, CD163, CSF1R,* and *CD68*) compartment to annotate them. In addition to manual annotation, we used the MatchScore2 package (version 0.1.0)[50] which compares the marker gene list with a reference dataset (in this case our dataset) and assigns a value (Jaccard index) to each comparison. High Jaccard indexes indicate high similarity between clusters.

Gene-sets from microarray analysis of stimulated macrophages were obtained from the analysis of GSE121825, GSE155719, GSE156921, GSE161774 and GSE94608 datasets[18,19,51,52]. Using these gene-sets, we calculated the average expression levels of each cluster on our dataset at single cell level, using the function AddModuleScore of the Seurat R Package (version 4.1.0).

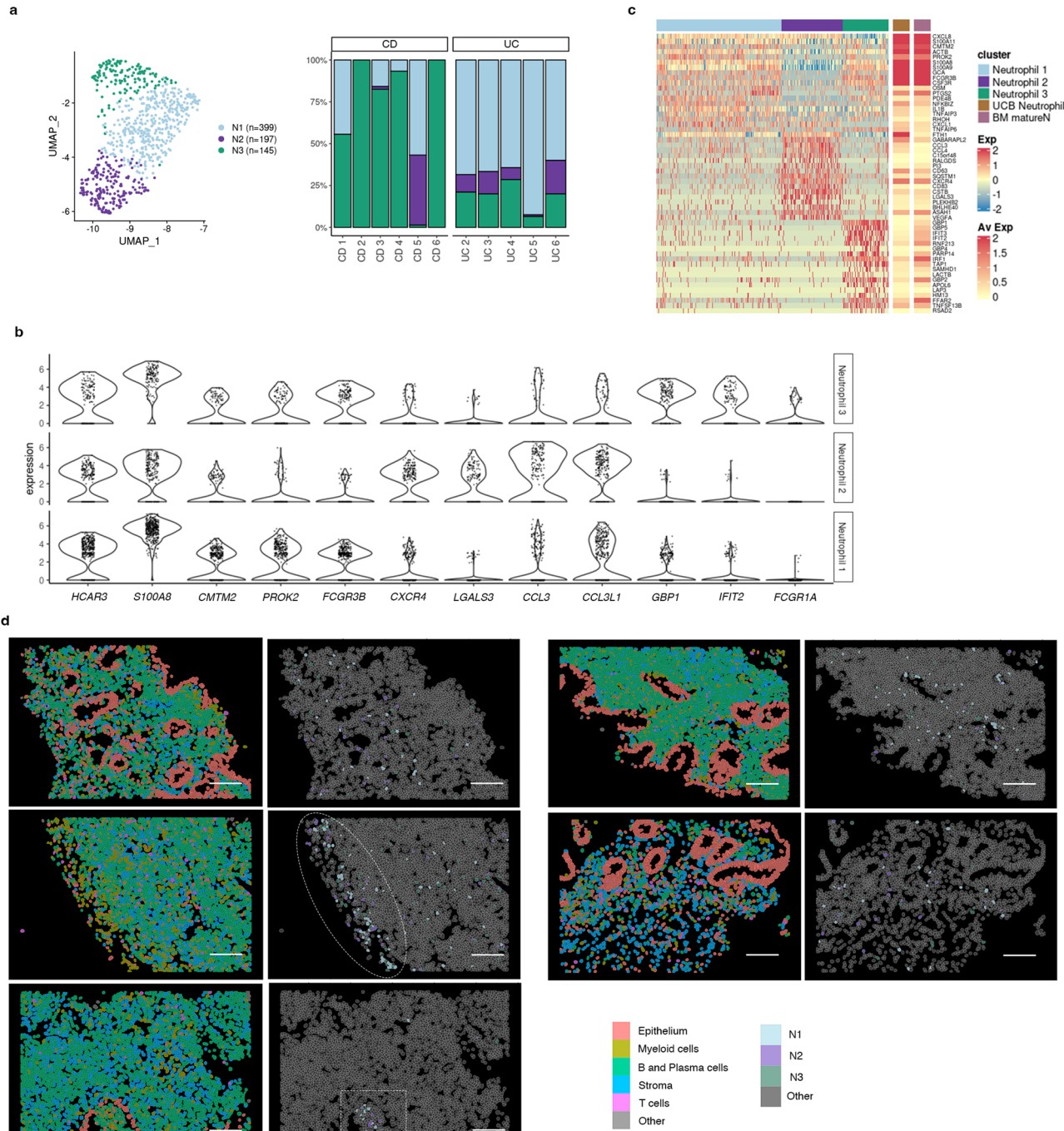

**Fig. 6 | Analysis of the heterogeneity of neutrophil populations in inflammatory bowel disease (IBD) colonic mucosa. a** UMAP showing the three neutrophil subsets/states (N1, N2, N3) observed in IBD samples by scRNA-seq analysis. Barplot representing the proportions of each neutrophil subset across health and IBD. **b** Violin plots visualizing the expression (x-axis) of marker genes common and specific for all three neutrophil populations (y-axis). **c** Heat map showing the average normalized and scaled expression of differentially expressed genes in all three neutrophil subsets. Average expression of these genes on neutrophils from cord blood and bone marrow mature neutrophils is shown on the far right (Xie X. et al. 2021, Zhao Y. et al. 2019*). **d** Representative CosMx SMI images of IBD inflamed tissue showing the spatial location of N1, N2 and N3 neutrophil subsets. Images correspond to 4 independent patients, 1 CD (showing 2 different Fields of View (FoV)) and 3 UC (1 FoV each). Circle shows the surface of an ulcer, and a square shape is used to indicate a crypt abscess. Source data are provided as a Source Data file. IBD, inflammatory bowel disease. CD, Crohn's disease. UC, ulcerative colitis.

## CosMx™ spatial molecular imaging (SMI)

CosMx™ technology [CosMx™ Human Universal Cell Characterization RNA Panel (1000-plex) + 20 custom genes; Nanostring, USA] was applied to 9 FFPE samples: 3 non IBD healthy controls, 3 CD and 3 UC with a mean of 19 fields of view (FoVs) per sample (171 total FoVs). The mean area explored was 1153 mm² per sample. The following cell surface markers were used for morphology visualization: B2M/CD298, PanCK, CD45, CD3 antibodies and DAPI.

**CosMx™ spatial molecular imager (SMI) sample preparation.** FFPE tissue sections were prepared for CosMx™ SMI profiling as described in ref. [15]. Briefly, five-micron tissue sections mounted on VWR

Superfrost Plus Micro slides (cat# 48311-703) were baked overnight at 60 °C, followed by preparation for in-situ hybridization (ISH) on the Leica Bond RXm system by deparaffinization and heat-induced epitope retrieval (HIER) at 100 °C for 15 min using ER2 epitope retrieval buffer (Leica Biosystems product, Tris/EDTA-based, pH 9.0). After HIER, tissue sections were digested with 3 μg/ml Proteinase K diluted in ACD Protease Plus at 40 °C for 30 min.

Tissue sections were then washed twice with diethyl pyrocarbonate (DEPC)-treated water (DEPC H2O) and incubated in 0.00075% fiducials (Bangs Laboratory) in 2X saline sodium citrate, 0.001% Tween-20 (SSCT solution) for 5 min at room temperature in the dark. Excess fiducials were rinsed from the slides with 1X PBS, then tissue sections were fixed with 10% neutral buffered formalin (NBF) for 5 min at room temperature. Fixed samples were rinsed twice with Tris-glycine buffer (0.1 M glycine, 0.1 M Tris-base in DEPC H2O) and once with 1X PBS for 5 min each before blocking with 100 mM N-succinimidyl (acetylthio) acetate (NHS-acetate, Thermo Fisher Scientific) in NHS-acetate buffer (0.1 M NaP, 0.1% Tween pH 8 in DEPC H2O) for 15 min at room temperature. The sections were then rinsed with 2X saline sodium citrate (SSC) for 5 min and an Adhesive SecureSeal Hybridization Chamber (Grace Bio-Labs) was placed over the tissue.

NanoString® ISH probes were prepared by incubation at 95 °C for 2 min and placed on ice, and the ISH probe mix (1 nM 980 plex ISH probe, 10 nM Attenuation probes, 1 nM SMI-0006 custom, 1X Buffer R, 0.1 U/μL SUPERase•In™ [Thermo Fisher Scientific] in DEPC H2O) was pipetted into the hybridization chamber. The chamber was sealed to prevent evaporation, and hybridization was performed at 37 °C overnight. Tissue sections were washed twice in 50% formamide (VWR) in 2X SSC at 37 °C for 25 min, washed twice with 2X SSC for 2 min at room temperature, and blocked with 100 mM NHS-acetate in the dark for 15 min. In preparation for loading onto the CosMx SMI instrument, a custom-made flow cell was affixed to the slide.

**CosMx Spatial Molecular Imager (SMI) instrument run.** RNA target readout on the CosMx SMI instrument was performed as described in Ref. 53. Briefly, the assembled flow cell was loaded onto the instrument and Reporter Wash Buffer was flowed to remove air bubbles. A preview scan of the entire flow cell was taken, and 15–25 fields of view (FoVs) were placed on the tissue to match regions of interest identified by H&E staining of an adjacent serial section. RNA readout began by flowing 100 μl of Reporter Pool 1 into the flow cell and incubation for 15 min. Reporter Wash Buffer (1 mL) was flowed to wash unbound reporter probes, and Imaging Buffer was added to the flow cell for imaging. Nine Z-stack images (0.8 μm step size) for each FoV were acquired, and photocleavable linkers on the fluorophores of the reporter probes were released by UV illumination and washed with Strip Wash buffer. The fluidic and imaging procedure was repeated for the 16 reporter pools, and the 16 rounds of reporter hybridization-imaging were repeated multiple times to increase RNA detection sensitivity.

After RNA readout, the tissue samples were incubated with a 4-fluorophore-conjugated antibody cocktail against CD298/B2M (488 nm), PanCK (532 nm), CD45 (594 nm), and CD3 (647 nm) proteins and DAPI stain in the CosMx™ SMI instrument for 2 h. After unbound antibodies and DAPI stain were washed with Reporter Wash Buffer, Imaging Buffer was added to the flow cell and nine Z-stack images for the 5 channels (4 antibodies and DAPI) were captured.

**Segmentation and quality control.** Images were segmented to obtain cell boundaries, assign transcripts at the cell-level, and obtain a transcript by cell count matrix[53]. Cells with an average negative control count greater than 0.5 and less than 20 detected features were filtered out. After quality control, a mean of 95,3% of cells across samples and FoVs was retained, corresponding to ~46,160 cells per sample on average.

**Preprocessing and feature selection.** Following library size normalization and log-transformation using logNormCounts (scater package, version 1.22.0)[54], highly variable genes (HVGs) were identified with modelGeneVar (scran package, version 1.22.1)[55], blocking by sample (i.e., variance modelling is performed per sample and statistics combined across samples), and selecting for genes with a positive biological variance component. For comparability with the lowest-coverage sample, per-sample scaling normalization with multiBatchNorm from the batchelor package (version 1.10.0 was applied[56].

**Integration and dimension reduction.** Next, Principal Component Analysis (PCA) was computed on highly variable genes (HVGs). Upon inspection of the corresponding elbow plot (% variance explained *vs.* # of PCs), we selected the first 30 PCs as input for integration using harmony[47], and for dimensionality reduction via uniform manifold approximation and projection (UMAP)[46].

**Cell type annotation.** To identify subpopulation markers, we ran findMarkers (scran package, version 1.22.1)[55], blocking by sample, considering HVGs only, and testing for positive log-fold changes. The 100 top ranked genes were selected and passed to SingleR (version 1.8.1)[57] along with the reference scRNA-seq dataset's count matrix and subpopulation assignments, for label transfer. To obtain higher-resolution annotations, we grouped cells into 5 biological compartments according to their lower-resolution label (namely: epithelial, myeloid, plasma, stroma, and T cells) and, for each compartment separately, reperformed marker detection and label transfer as described above.

**Co-localization analysis.** To quantify how pairs of cell types co-localize in a given sample and FoV, we computed, for every sample, FoV, and pair of cell types, two-dimensional kernel density estimates (KDEs) of cell coordinates using kde. Within each sample-FoV, estimation was performed over a rectangular window according to the boundary coordinates of cells from a given pair of types, and the Pearson correlation coefficient between KDEs was computed. Only comparisons with more than 10 cells from both types were taken into consideration.

**Mucosal-surface distance calculation.** To determine the distance of each myeloid cell type from the mucosal surface, we drew an imaginary line at the surface epithelium and calculated the Euclidean distance between the centroid of each cell and the nearest point on the imaginary line. Only myeloid subsets with at least 10 cells detected were included in the analysis. The minimum, mean, median, maximum, and standard deviation of distances were then calculated for each cluster and health group based on the FOVs in Fig. 2e. The summarized data can be found in Supplementary Table 3.

**Correlation abundance analysis.** The abundances of each cell type/cluster were calculated by aggregating the cell counts from each patient. Pearson correlation coefficients and corresponding p-values were calculated using the rcorr function from the HmiscR package (version 4.6-0) to compare the abundances of each cell type/cluster between patients. The resulting correlation coefficients were visualized using corrplot from the corrplot package (version 0.92)[58]. To account for multiple testing, the p-values were adjusted using the false discovery rate (Benjamini-Hochberg procedure).

**Granulocytes isolation from blood**
Human granulocytes were isolated from blood (n = 6 donors) by a double density gradient. In brief, diluted blood in PBS (1/2) was

layered over Lymphoprep™ (1.077 g/ml) that was layered over denser Polymorphprep™ (1.113 g/ml). The double gradient was then centrifuged at 500 g for 30 min obtaining 2 separated cell layers. The lower layer containing the granulocytes was collected, washed and red blood cells were lysed using a commercial lysis buffer (BioLegend). Purity of granulocytes achieved with this method is > 95%.

## Flow cytometry
Blood granulocytes and single-cell suspensions obtained from digested biopsies of heathy donors and IBD patients were stained with the following antibodies CD66b PE-Cy7 (#305116 clone G10F5, 1/20 dilution, BioLegend), CD16 PerCP (#302028 clone 3G8, 1/20 dilution, BioLegend), CD62L APC-Cy7 (#304814 clone DREG-56, 1/20 dilution, BioLegend), CD69 BV421 (#310930 clone FN50, 1/20 dilution, BioLegend), CD193 FITC (#310720 clone 5E8, 1/20 dilution, BioLegend), CD63 FITC (#353005 clone H5C6, 1/20 dilution, BioLegend) and CXCR4 PE (# FAB170P clone 12G5, 1/10 dilution, R&D systems) and Zombie Aqua Fixable Viability Kit 1/1000 (#423101, BioLegend) for death cells. Cells were fixed using BD Stabilizing Fixative [BD], acquired using a BD FACSCanto II flow cytometer (BD) and analyzed with FlowJO software (version 10.6.1, BD).

## Organoid cultures from human colonic crypts and RT-PCR
Epithelial 3D organoid cultures were generated using surgical samples from adult healthy donors ($n = 13$). Briefly, intestinal crypts were mechanically isolated after 8 mM EDTA incubation for 45 min at 4 °C under mild agitation. Crypts were embedded in Matrigel (BD Biosciences, CA, USA) (approximately 90 crypts/25 μl Matrigel) and overlaid with organoid growth medium[59]. Ex vivo organoids embedded in Matrigel (BD Biosciences, CA, USA) were passaged for further expansion approximately every 5–6 days (1:5 dilution on average) using a dispase-based cell dissociation protocol. After 1–3 passages, 4–5-day old organoid samples were stimulated with neuregulin 1 (R&D, Germany) at 1 μg/mL for 48 h in growth medium[59] depleted of human Epidermal Growth Factor (EGF; ThermoFisher), LY-2157299 (Axon MedChem) and SB-202190 (Sigma-Aldrich). Organoid samples were then harvested in Trizol (Ambion, Thermo Fisher Scientific, USA) and RNA isolated using GeneJET RNA Cleanup and Concentration Micro Kit (Thermo Fisher Scientific, USA). Total RNA was transcribed to cDNA using reverse transcriptase (High Capacity cDNA Archive RT kit, Applied Biosystems, Carlsbad, CA, USA) Quantitative real-time PCR (qPCR) was performed in an ABI PRISM 7500 Fast RT-PCR System (Applied Biosystems) using predesigned TaqMan Assays (Applied Biosystems) (*OLFM4* Hs00197437_m1, *LGR*5 Hs00173664_m1 and *MKI67* Hs00606991_m1).

## Bulk RNA-seq analysis of human colonic biopsies
Colonic biopsies were taken during routine endoscopy from healthy controls ($n = 8$), active UC ($n = 26$) and active CD ($n = 22$) patients. Samples were placed in RNAlater RNA Stabilization Reagent (QIAGEN, Hilden, Germany) and stored at −80 °C until RNA isolation. RNA was isolated using an RNeasy Kit (QIAGEN, Hilden, Germany) according to the manufacturer's instructions and sequenced for bulk RNA-seq. Barcoded RNAseq libraries were prepared from total RNA using a TruSeq stranded mRNA kit (Illumina, San Diego, CA, USA) according to the manufacturer's instructions. Libraries were subjected to single-end sequencing (50 bp) on a HighSeq-3000 platform (Illumina, CA, USA) at the Translational Medicine and Genomics group (Boehringer-Ingelheim GmbH & Co, Biberach, Germany). Quality filtering and adapter trimming was performed using Skewer version 2.2.8 Reads were mapped against the human reference genome using the STAR aligner version 2.5.2a. The genome used was GRCh38.p10, and gene annotation was based on Gencode version 27 (EMBL-EBI, Hinxton, UK). Read counts per gene were obtained using

RSEM version 1.2.31 and the Ensembl GTF annotation file (EMBL-EBI, Hinxton, UK). Analyses were performed using the R (version 3.2.3) statistics package.

## Statistical analysis
Graphs of bulk RNA-seq, qPCR from organoids and flow cytometry results and statistical analysis were performed using Graphpad Prism 9.0 (Graphpad Software, CA, USA). Differences between groups in the bulk RNA-seq data were tested using Ordinary one-way ANOVA test and correcting for multiple comparisons by controlling the false discovery rate [two-stage linear step-up procedure of Benjamini, Krieger and Yekutieli]. For organoid results, one-sample t test was performed. Differences between groups in flow cytometry data were tested using the non-parametric Kruskal–Wallis test and correcting for multiple comparisons by controlling the false discovery rate (two-stage linear step-up procedure of Benjamini, Krieger and Yekutieli).

## Immunohistochemistry (IHC) and immunofluorescence (IF)
Immunostaining of tissue sections of healthy donors ($n = 12$ for CD68, CD209, $n = 9$ for MPO, MBP and $n = 3$ for olfactomedi4) and IBD patients (UC $n = 3$ for OLFM4, $n = 6$ for CD68, MPO, MBP; CD $n = 3$ for olfactomedin, $n = 5$ for MPO, MBP) was performed using commercially available antibodies (anti-rabbit CD68 1/500 IHC and 1/200 IF (#HPA048982 Sigma, MO, USA), anti-mouse CD209 1/500 IHC and 1/200 IF (#SC-65740 clone DC28, Santa Cruz Biotechnologies, TX, USA), anti-rabbit OLFM4 1/100 (#14369 S Cell Signaling, MA, USA), anti-rabbit MPO 1/3000 (#HPA021147 Sigma, MO, USA), anti-mouse MBP 1/100 (#CBL419 clone BMK13, Sigma, MO, USA)). Deparaffinization, rehydration and epitope retrieval of the samples was performed using PT Link (Agilent, CA, USA) using Envision Flex Target Retrieval Solution Low pH (Dako, Germany). Blocking of the samples was performed using horse or goat serum (10–20%) (Vector, NY, USA) in a PBS + 0,5% BSA solution. Secondary antibodies were used for IHC (goat anti-rabbit 1/200 and horse anti-mouse 1/200 (Vector, NY, USA) and IF (goat anti-rabbit AF488 1/400 and goat anti-mouse Cy3 1/400). For IHC the immunoperoxidase detection system was used (Vector, NY, USA). DAPI (Thermo Fisher Scientific, USA) counterstaining was performed on IF samples. Image acquisition was performed on a Nikon Ti microscope (Japan) using Nis-Elements Basic Research Software (version 5.30.05). Immunofluorescence composite was performed using ImageJ software (version 1.53t).

## Single molecule RNA in situ hybridization (ISH)
ISH was carried out on FFPE tissues from HC ($n = 3$) and IBD patients (UC $n = 6$, CD $n = 3$) fixed in 10% neutral buffered formalin for 48 h. RNA probes for *NRG1* (cat#311181) and *OLFM4* (cat#311041) and RNAscope 2.5 HD assay – Red kit (Biotechne, CO, USA) were used according to the manufacturer's instructions. Pre-warmed probes were added to the slides and incubated in the HybEZ oven (Biotechne, CO, USA) for 2 h at 40 °C. After a 6-step signal amplification, for double IHC and ISH samples, IHC protocol was followed. Tissue sections were counterstained with Gill's hematoxylin. Slides were mounted with ECOmount mounting medium (Biocare medical, CA, USA) and photographed using Nikon Ti microscope (Japan) and Nis-Elements Basic Research Software.

## Reporting summary
Further information on research design is available in the Nature Portfolio Reporting Summary linked to this article.

# Data availability
ScRNA-seq, bulk RNA-seq and CosMx SMI raw data generated in this study have been deposited in the GEO Omnibus database under accession code GSE214695, GSE235236 and GSE234713, respectively.

Processed scRNA-seq and CosMx data can be explored at https://servidor2-ciberehd.upc.es/external/garrido/methods1/. Source data are provided with this paper.

## Code availability

Full code for scRNAseq analysis is available at https://github.com/ibd-bcn/ibd-bcn_single_cell and our webpage https://servidor2-ciberehd.upc.es/external/garrido/methods2/. Full code for analysis of CosMx SMI data can be found at https://github.com/HelenaLC/CosMx-SMI-IBD.

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

## Acknowledgements
We thank the Pathology Departments at Hospital Clinic Barcelona and the Biobank Facility and Flow Cytometry unit at the Institut d'Investigacions Biomèdiques August Pi i Sunyer (IDIBAPS) for providing us with the samples required to conduct this study and for the technical help, and all patients for their selfless participation. Patrick Baum from Boehringer-Ingelheim for the sequencing of the samples. Cátia Moutinho for her help and guidance in single-cell experiments. We also thank members of our group Agnès Fernandez-Clotet for support with patient recruitment, Ángela Sanzo and Marc Moro for support with data analysis and Joe Moore and Alicia Garrido for editorial assistance. We are also indebted to Maite Rodrigo-Calvo from the Pathology Department at Hospital Clínic Barcelona, for help with patient evaluation.

## Author contributions
A.G.T. conceptualized the results of the manuscript, performed experiments, interpreted data, created figures, and wrote manuscript draft. A.M.C. performed computational analysis of scRNAseq and CosMx SMI, generated figures and wrote computational methods. M.V. performed and supervised experiments, wrote methods, and reviewed and edited manuscript. I.D. performed experiments and reviewed and edited manuscript. A.R. performed computational analysis of scRNA-seq. H.L.C. performed CosMx SMI computational analysis. V.G. helped with the conceptualization of the manuscript, reviewed and edited manuscript. E.M.-A., I.A.-T., and Mi.E. performed experiments. D.A. analyzed bulk RNA-seq data. E.K., Y.K., and M.L. performed CosMx experiments. S.V. sequenced bulk RNAseq of human biopsies. D.M. and G.C. performed 10x Genomics experiments. A.M.-C., Ma.E., I.O., M.C.M. and E.R. provide human samples and clinical information. J.P. acquired funding and critically reviewed the manuscript. E.M. supervised, advised, and contributed to computational analysis and reviewed the manuscript. H.H. provided technical and computational support, reviewed the manuscript. A.S. acquired funding, designed, and supervised the project, and wrote the manuscript.

## Funding
This work was funded by grant PID2021-123918OB-I00 from MCIN/AEI/10.13039/501100011033 and co-funded by "FEDER A way to make Europe". AM-C, VG and ID were funded by grant Grant #2008-04050 from The Leona and Harry B. Helmsley Charitable Trust. EM-A is funded by grant RH042155 (RTI2018-096946-B-I00) from Ministerio de Ciencia e Innovacion. IA-T is funded by grant 831434-2 (CE_IMI2-2018-14 call). HH received support for the project PID2020-115439GB-I00- funded by MCIN/AEI/10.13039/501100011033. This publication is part of a project that has received funding from the Innovative Medicines Initiative 2 Joint Undertaking under grant agreement No 831434. This project also has received funding from the European Union's Horizon 2020 research and innovation program under grant agreement No 848028.

## Competing interests
H.H. is co-founder of Omniscope, a scientific advisory board member of MiRXES and consultant to Moderna. A.S. is the recipient of research grants from Roche-Genentech, Abbvie, GSK, Scipher Medicine, Pfizer, Alimentiv, Inc, Boehringer Ingelheim and Agomab; receives consulting fees from Genentech, GSK, Pfizer, HotSpot Therapeutics, Alimentiv, Origo Biopharma, Deep Track Capital, Great Point Partners and Boxer Capital; and is on the advisory boards of BioMAdvanced Diagnostics, Goodgut and Orikine. MaE. has received support for conference attendance and research support from Abbvie, Biogen, Faes Farma, Ferring, Jannsen, MSD, Pfizer, Takeda, and Tillotts. J.P. received financial support for research from AbbVie and Pfizer; consultancy fees/honorarium from AbbVie, Arena, Athos, Atomwise, Boehringer Ingelheim, Celgene, Celltrion, Ferring, Galapagos, Genentech/Roche, GlaxoSmithKline, Janssen, Mirum, Morphic, Nestlé, Origo, Pandion, Pfizer, Progenity, Protagonist, Revolo, Robarts, Takeda, Theravance and Wasserman; reports payment for lectures including service on speaker bureau from Abbott, Ferring, Janssen, Pfizer and Takeda; and reports payment for development of educational presentations from Abbott, Janssen, Pfizer Roche and Takeda. A.M-C has received financial support for conference attendance, educational activities, and research support from Abbvie, Biogen, Ferring, Jannsen, MSD, Takeda, Dr. Falk Pharma and Tillotts. E.K., Y.K., and M.L. are current/former employees and shareholders of NanoString Technologies. S.V. is a current employee of Boehringer-Ingelheim Pharmaceuticals. The remaining authors declare no competing interests.

## Additional information

[1]Inflammatory Bowel Disease Unit, Institut d'Investigacions Biomèdiques August Pi I Sunyer (IDIBAPS), Hospital Clínic, Barcelona, Spain. [2]Centro de Investigación Biomédica en Red de Enfermedades Hepáticas y Digestivas (CIBEREHD), Barcelona, Spain. [3]Josep Carreras Leukaemia Research Institute (IJC), Badalona, Spain. [4]Department of Molecular Life Sciences, University of Zurich, Switzerland. SIB Swiss Institute of Bioinformatics, Zurich, Switzerland. [5]Centro de Investigaciones Biológicas, Consejo Superior de Investigaciones Científicas (CSIC), Madrid, Spain. [6]NanoString Technologies, Seattle, WA, USA. [7]Translational Medicine and Clinical Pharmacology, Boehringer-Ingelheim Pharmaceuticals Inc, Ridgefield, CT, USA. [8]CNAG-CRG, Centre for Genomic Regulation (CRG), Barcelona Institute of Science and Technology (BIST), Barcelona, Spain. [9]Department of Gastroenterology, Hospital Universitari Mútua Terrassa, Universitat de Barcelona, Terrassa, Spain. [10]Universitat Pompeu Fabra (UPF), Barcelona, Spain. [11]These authors contributed equally: Elisabetta Mereu, Holger Heyn. ✉e-mail: asalas1@recerca.clinic.cat

