## [Peer Review File · Nature Communications]

REVIEWER COMMENTS

Reviewer #1 (Remarks to the Author):

Garrido-Trigo et al. present an analysis of colonic tissue cells in IBD, Crohn's and controls at both the single cell RNAseq level, as well as spatially, using CosMx. They explore in depth the subtypes of myeloid cells, found to be most heterogeneous among patients, and identify three sets of macrophages and three sets of neutrophils, differing in cellular identity and localization within the tissue. They further identify NRG1+ subsets of macrophages that are spatially associated with fibroblasts, suggesting their functional induction of polarization. The data forms an excellent resource to explore molecular signatures of diverse cell populations in IBD. I have a few generally minor comments to be addressed in a revision:

- Figure 2e – It would be interesting to add a systematic analysis of the distance of each of the cell types from the mucosal surface.
- Are there cell types that correlate/anticorrelate in abundance among different patients? It would be good to add such analysis, potentially exposing homeostatic interactions between diverse cell types.
- All samples were from the colon, ileal Crohn's disease is not examined, the authors should mention this as a limitation of the study in the Discussion.
- Control samples were taken from the sigmoid, are there IBD biopsies from the sigmoid? If there are it would be important to demonstrate that differences in cell identities are not merely difference in the colonic segment analyzed.
- Olfm4 is a target of both Notch and Tnfa (PMID 33490652), the authors should examine the change in other Notch and Tnfa targets.
- Figure 5a – add X axis label.
- All image panels – add scale bars.
- Figure 6d – the colors of Epithelium and N1 are too similar, please change.
- Discussion – lines 650-652 – this data is not shown, either show or remove.
- Discussion – line 596-600 – the authors discuss at length the serotonin effect but this only appears in Extended Data Figure 5 with no mention in the Results, please describe this more explicitly in the results section.

Reviewer #2 (Remarks to the Author):

Garrido-Trigo et al. report on Macrophage and neutrophil heterogeneity at single-cell spatial resolution in inflammatory bowel disease. This is a relevant study because it adds information on unreported cell populations, in particular neutrophils.

Major points relate to cell clustering, batch effects, and differential harmonisation.

Epithelial cells required a less stringent filter of 65% of counts aligned to the mitochondrial genes for quality control.

> can you explain why this is? Are the epithelial cells dying and therefore you accept lower quality?

The cluster-recluster strategy outlined in the methods is not clear- is this partially non-supervised and partially supervised?

>"Each main cell type was re-processed starting from FindVariableFeatures using the same procedure as for the whole dataset we further removed CD3, C1QA, DERL3, or MS4A1 expressing cells from Epithelial cells; CD3, C1QA, or EPCAM-expressing cells from the B cells; CD3, C1QA, DERL3, MS4A1 or EPCAM-expressing cells from the Stromal cells; DERL3, MS4A1, C1QA, EPCAM, or CD79A-expressing cells from the Lymphoid cells; and CD3, THY1, DERL3 or MS4A1-expressing cells from the Myeloid cells. ...

Were these contaminations regarded as " doublets " ?

Figure 1 c and d > it is not clear what the sub-segmentation in the bar chart represents

Neutrophil 1 largely driven by 2 samples, Neutrophil2 is driven by a single CD sample

> are there any histological differences?

Extended Data Figure 2. Is confusing, Lit

The authors observed batch effect by sample in the UMAPs of Lymphoid (tcells), B and plasmacells (plasmas), and Stromal cells (stroma) objects while re-processing for cell type identification. For those cell types, we ran Harmony 5(version 0.1.0) to correct for this effect.

> why would the myeloid cells be less affected by batch effects? This is not convincing

The M&M section Annotation of cells contains a wordy section of supplemental results - a table would be easier to process

Figure 2 is very difficult to follow because the colors are overlapping > suggest to revise

"Subsets, such as inflammatory fibroblasts, neutrophils, or inflammatory M1 macrophages ... were absent in HC"

A total of 46,700 cells across 18 samples suggests that some of the cell types not identified in HC might be due to low numbers . Therefore some of the results might be biased by the low cell counts.

Reviewer #3 (Remarks to the Author):

In this manuscript by Garrido-Trigo et.al., the authors utilized a combination of scRNA-seq and spatial imaging to identify some interesting phenotypes in macrophages and neutrophils in intestinal tissues that are associated with inflammation. Notable findings include the identification of some novel markers for interesting macrophage populations, as well as the association, based on spatial imaging with inflammatory fibroblasts. Certainly, the results support the notion that macrophage and neutrophil populations are more complexed than previously appreciated.

However, the main weakness in this study is that the authors do not relate the phenotype of the cellular populations that they observed with the clinical phenotypes of the patients. As the authors note, IBD is marked by perplexing inter-individual variation in clinical phenotype, pathology and response to treatment. The application of single cell and other new technologies has the potential to uncover the basis of inter-patient heterogeneity. But in this study, the authors do not relate the interesting phenotypes they observed with clinical outcomes, or even phenotypes. Hence the overall impact of the study is more limited. Perhaps with some meta-analysis of other datasets, or additional samples, the authors may be able to relate the cellular phenotypes that they have identified with some clinical features of IBD, which may improve the overall manuscript.

REVIEWER COMMENTS

R: *We thank the reviewers and the editorial team for their constructive comments and the time devoted to reviewing our manuscript.*

Reviewer #1 (Remarks to the Author):

Garrido-Trigo et al. present an analysis of colonic tissue cells in IBD, Crohn's and controls at both the single cell RNAseq level, as well as spatially, using CosMx. They explore in depth the subtypes of myeloid cells, found to be most heterogeneous among patients, and identify three sets of macrophages and three sets of neutrophils, differing in cellular identity and localization within the tissue. They further identify NRG1+ subsets of macrophages that are spatially associated with fibroblasts, suggesting their functional induction of polarization. The data forms an excellent resource to explore molecular signatures of diverse cell populations in IBD. I have a few generally minor comments to be addressed in a revision:

- Figure 2e – It would be interesting to add a systematic analysis of the distance of each of the cell types from the mucosal surface.

R: To assess the distance of each myeloid cell type to the mucosal surface, we draw an imaginary line at the surface epithelium and systematically measured the mean and average median distance for all FOVs in Figure 2.e. Myeloid subsets with at least 10 cells detected were included in this analysis. In healthy samples (HC) resident macrophages were the closest to the mucosal surface, whereas mast cells and DCs tended to localize in deeper mucosal sections. Again, we observed shifts in the inflamed sections. M1 macrophages and N1/N3 neutrophils were closest to the mucosal surface (localizing in superficial ulcers), whereas N2 (CXCR4) neutrophils were found in deeper mucosal areas, in agreement with the colocalization analysis described below (new extended Figure 1d). Indeed, spatial analysis shows that N2 neutrophils are predominantly found in a neighborhood together with submucosal S3 fibroblasts and myofibroblasts at the muscularis mucosa.

This is now described in the Results section (Page 17) and the data shown in Supplementary Table 5.

- Are there cell types that correlate/anticorrelate in abundance among different patients? It would be good to add such analysis, potentially exposing homeostatic interactions between diverse cell types.

R: Following the reviewer's suggestion we have performed correlation analysis on the abundances of cell types using the scRNA-seq data. In addition, a correlation of the spatial cell distribution was calculated among patients using the CosMx SMI dataset. The methodology used for this analysis is described in Supplementary Methods. Given the marked differences in cell composition, as well as the tissue architecture between healthy and inflamed colon, we conducted the analysis of healthy and inflamed (IBD) samples separately (Extended Figure 1c). Overall, we observed the following correlations in total abundances: In healthy colon there were positive trends in cell type abundances (scRNA-seq data); however, none were significant (adjusted p value). Within IBD samples, significant positive correlations were found between Cycling, GC and memory B cells. Similarly, the abundances of epithelial cell subsets tended to correlate with each other, while structural fibroblasts, including S1 (lamina propria), S2 (peri-

cryptal), S3 (submucosal) and myofibroblasts (muscularis mucosa), were positively correlated. Remarkably, we found that Glial cells correlated significantly with BEST4OTP2, Colonocytes and the Ribhi epithelium, as well as M2 macrophages and S1 (lamina propria fibroblasts). While not to a significant degree, the abundance of glial cells anti-correlated with most inflammatory cells including neutrophils, inflammatory fibroblasts, M1 macrophages and inflammatory monocytes. This would suggest that the abundance of glial cell is associated with a healthy intestinal architecture, while inflammation may lead to the loss of this cell type.

Regarding the spatial co-localization of cell-types, we observed important differences between HC and IBD samples, in agreement with the marked architectural and compositional changes seen in inflamed tissues. Based on their spatial correlation, we defined different neighborhoods (Extended Figure 1d). In HC, cells known to form lymphoid tissues in the intestine (DCs, different types of T cells, FRCs, cycling cells, GC B cells, B cell and naïve B cells) were spatially co-localized across samples (Neighborhood I in HC). At least 3 other structural clusters were found, including a top lamina propria formed by Plasma cells, BEST4 OTOP2 enterocytes, S1 and S2 fibroblasts (Neighborhood II), a lower crypt cluster (including progenitor and cycling epithelium, Neighborhood III) and a submucosa area (containing S3 fibroblasts and myofibroblasts, pericytes, endothelial cells, glia and mast cells; Neighborhood IV).

Remarkably, in inflamed samples, B cell subsets continued to localize with T cells (Neighborhoods IV and V), but now also included T cell CCL20 and IDA macrophages. Neighborhood II represented inflammatory foci (M1 macrophages, N3 neutrophils, inflammatory fibroblasts, and inflammatory monocytes). Neighborhood III contained upper crypt cells (BEST4OTOP2, colonocytes, M2 and S2b fibroblasts), while Neighborhood I was a highly heterogeneous neighborhood containing all Plasma cells, “submucosal” cell types (S3, Myofibroblast), lamina propria cell types (Mast, Endothelium, Pericytes, S1, Fibroblasts, Glia, S2a), epithelium (Enteroendocrine, Tuft cells, Goblet, Secretory progenitor, Paneth-like, Cycling TA and Epithelium Ribhi), in addition to N1 and N2 neutrophils.

We have added new analyses to the Results section on page 8 and in Extended Figures 1c-d.

- All samples were from the colon, ileal Crohn’s disease is not examined, the authors should mention this as a limitation of the study in the Discussion.

R: Indeed, we focused exclusively on colonic disease with the aim of comparing colon samples from healthy, UC and colonic CD patients. While ileal disease is very common in CD, the transcriptomes of ileum and the colon are markedly different based on bulk RNA analysis (PMID: 34098982), which would have affected any comparison with colonic samples (in UC). Nonetheless, analysis of an ileal cohort (Martin et al.) is included in Extended data Figure 4 and shows the similarities in the myeloid compartment compared to our colonic cohort (mentioned now in the Discussion, page 51). In addition, we now comment on this limitation in the Discussion (Page 54).

- Control samples were taken from the sigmoid, are there IBD biopsies from the sigmoid? If there are it would be important to demonstrate that differences in cell identities are not merely difference in the colonic segment analyzed.

R: This is an important point. As we discussed in our previous response, most differences are seen when comparing the ileum to the colon. Nonetheless, even within the colon some discrepancies in cellular composition may occur. To acknowledge this point, we included in Supplementary Table 1 the specific location of the samples used in our

analysis. Three out of six UC samples, and four out of six of the CD samples, were taken from the sigmoid colon, suggesting that colonic location may not explain the differences between different cell identities. Unfortunately, due to the low number of samples per group, we could not perform a comparison based on disease location at this time.

- Olfm4 is a target of both Notch and Tnfa (PMID 33490652), the authors should examine the change in other Notch and Tnfa targets.

R: We thank the reviewer for pointing this out. Indeed, *OLFM4* can be a target of Notch and TNF (NFKB) signaling, both important pathways in the homeostasis of the epithelium in health and IBD. We looked at the expression of the Notch target genes *HES1*, *HEY1* and *HEY2*, *RHOV*, *STC1*, *SNAI2*, *UBD* and *KRT13* (PMID: 30395194), and Notch receptors (*NOTCH1-4*) within the intestinal epithelium. Of all these genes, only the expression of *HES1* and *NOTCH1* was detectable within *OLFM4*⁺ cells (Extended Figure 7a). In contrast to *HES1*, which was expressed across epithelial cell clusters, *NOTCH1* was found predominantly within undifferentiated or progenitor cell types. Regarding TNF signaling, we investigated expression of the TNF receptors *TNFRSF1A* and *TNFRSF1B* (not shown), as well as target *TNFAIP3* (new Extended Figure 7a). In contrast to *NOTCH*-related genes, all three TNF-related genes were preferentially expressed within the differentiated colonocyte populations, including inflammatory colonocytes, in addition to secretory progenitors (new Extended Figure 7a). Based on this, we conclude that the *NOTCH* and the TNF signaling may be targeting at least partially different epithelial subsets or states. To confirm this hypothesis, we also determined the spatial localization of *NOTCH1* and *TNFRSF1A*, which were both included in the 1000-gene *CosMx* panel (new Extended Figure 7b). This confirmed the different localization along the intestinal crypt of the *NOTCH* and TNF receptors.

Based on these patterns, we suggest that the *NOTCH* pathway may be, at least in part, involved in driving *OLFM4* expression in the TA and progenitor epithelium in the healthy intestine. In contrast, in inflamed tissues, where both *OLFM4* and *NRG1* are highly expressed (but not *HES1* or *NOTCH1*), we suggest instead a role of neuregulin 1 in driving changes in *OLFM4* (comments have been included on pages 28 and 29). Nonetheless, further experiments will be needed to specifically address this point.

- Figure 5a – add X axis label.

R: The axis label has been added as the reviewer suggested.

- All image panels – add scale bars.

R: Scale bars have been added to the images.

- Figure 6d – the colors of Epithelium and N1 are too similar, please change.

R: Colors on the figure have been changed.

- Discussion – lines 650-652 – this data is not shown, either show or remove.

R: We have removed this line from the Discussion.

- Discussion – line 596-600 – the authors discuss at length the serotonin effect, but this only appears in Extended Data Figure 5 with no mention in the Results, please describe this more explicitly in the results section.

R: We fully agree with the reviewer that we did not provide sufficient data to support an *in vivo* role of serotonin as a potential driver of IDA macrophage differentiation. In addition, we do not detect any expression (at the mRNA level) of the serotonin receptors in intestinal M2 macrophages. Hence, we have revised our

comments on the potential role of serotonin in the Discussion. However, we have kept the data on the serotonin-induced profile in vitro as it remains of potential interest to understanding the origin of this population.

Reviewer #2 (Remarks to the Author):

Garrido-Trigo et al. report on Macrophage and neutrophil heterogeneity at single-cell spatial resolution in inflammatory bowel disease. This is a relevant study because it adds information on unreported cell populations, in particular neutrophils.

Major points relate to cell clustering, batch effects, and differential harmonisation. Epithelial cells required a less stringent filter of 65% of counts aligned to the mitochondrial genes for quality control. Can you explain why this is? Are the epithelial cells dying and therefore you accept lower quality?

R: We thank the reviewer for bringing up this point. Epithelial cells tend to die rapidly upon tissue digestion as they lose their cell-to-cell interactions. This preferentially affects the terminally differentiated compartment (upper crypt and surface epithelium), although it is observed across the epithelial compartment. Thus, the percentage of mitochondrial genes in epithelial cells is markedly increased compared to the other cell compartments. This effect is consistent with other reports (Smillie et al. Parikh et al.). We considered different approaches, including using the same percentage (25%) cutoff across cell types, but that led to losing important cell types. We found a compromise using a cutoff of 65% for the epithelial compartment, below which cells could be confidently annotated and assigned a profile. Nonetheless, the raw data provided with the publication will allow other investigators to establish their own criteria for filtering and exploring the data freely.

- The cluster-recluster strategy outlined in the methods is not clear- is this partially non-supervised and partially supervised?

R: We apologize for the lack of clarity in this section. We have reviewed and modified the Methods section (Data processing, page 2-3) to clarify this area. Indeed, our clustering approach is unsupervised. Nonetheless, the assignment of each cluster to a specific main cell type (epithelium, stroma, myeloid, etc) was done manually. We hope this clarifies any confusion.

- "Each main cell type was re-processed starting from FindVariableFeatures using the same procedure as for the whole dataset ... we further removed CD3, C1QA, DERL3, or MS4A1 expressing cells from Epithelial cells; CD3, C1QA, or EPCAM-expressing cells from the B cells; CD3, C1QA, DERL3, MS4A1 or EPCAM-expressing cells from the Stromal cells; DERL3, MS4A1, C1QA, EPCAM, or CD79A-expressing cells from the Lymphoid cells; and CD3, THY1, DERL3 or MS4A1-expressing cells from the Myeloid cells. ... Were these contaminations regarded as " doublets " ?

R: We have updated the Methods section to makes this section more clear to the reader. As mentioned, the revised version of the Methods section (page 3) now indicates those cells were removed based on the assumption that they doublets. We hope that this additional information better explains our approach.

- Figure 1 c and d > it is not clear what the sub-segmentation in the bar chart represents.

R: We thank the reviewer for pointing this out. We have now added an explanation to the Figure 1 legend. The sub-segmentation shows the contribution of each individual sample (patient).

- Neutrophil 1 largely driven by 2 samples, Neutrophil2 is driven by a single CD sample

> are there any histological differences?

R: Indeed, both the number of neutrophils and the proportion of each neutrophil subset was highly variable among patients. Patient CD5, as pointed out by the reviewer, contributes most N1 and N2 neutrophils (Extended Figure 1a). Interestingly, this patient also possessed the largest proportions of CXCL5_M1 macrophages and inflammatory fibroblasts compared to all other samples, suggesting that these cell types potentially play a role in driving neutrophil recruitment in this patient. While this is an interesting observation, it remains anecdotal and further exploration (involving additional data sets) is required to draw any robust conclusions.

Regarding the reviewer's question about histologic changes associated with specific neutrophil subsets, we did not observe any correlation between the composition of the neutrophil pool and histologic features. The largest histologic differences, as one would expect, were observed when comparing CD and UC patients. In IHC analysis (using an anti-MPO antibody), neutrophils were more abundantly found in UC samples, predominantly in the lamina propria, intraepithelial compartment and in crypt abscesses. This was associated with an increased epithelial loss in UC compared to CD, suggesting, at least in some patients, that increased neutrophil levels within crypts leads to crypt destruction. While we cannot fully resolve this question, we have added correlation analysis of cell types (Extended Fig 1c and 1d) and found that neutrophils (particularly N3) tend to localize with M1 macrophages within ulcerated areas, suggesting a relation between ulcer formation and neutrophil infiltration.

- Extended Data Figure 2. Is confusing, Lit

R: We apologize for the lack of clarity. We have simplified the annotation in panels a_iii and b_iii, and provided a better description of the contents in the Figure legend.

-The authors observed batch effect by sample in the UMAPs of Lymphoid (tcells), B and plasmacells (plasmas), and Stromal cells (stroma) objects while re-processing for cell type identification. For those cell types, we ran Harmony 5(version 0.1.0) to correct for this effect.

> why would the myeloid cells be less affected by batch effects? This is not convincing

R: As the reviewer points out, we did not correct for batch effect in the myeloid (or epithelial) compartment. We carefully evaluated each compartment by measuring the entropy within. As shown (Figure below), the entropy for Lymphoid (T cells), B and plasma cells (plasmas), and Stromal cells (stroma) was significantly lower than that of the myeloid compartment, suggesting a higher batch effect within those compartments.

We have now clarified this important point in the Methods section (page 3). While we agree that correction of the batch effect is a critical step in scRNA-seq data analysis, we tried to avoid the over-correction of samples since that can lead to loss of important patient-to-patient heterogeneity.

-The M&M section Annotation of cells contains a wordy section of supplemental results - a table would be easier to process.

R: Following the reviewer's suggestion, we have replaced this section with a table summarizing the annotation results in the Methods section (pages 3-4).

-Figure 2 is very difficult to follow because the colors are overlapping > suggest to revise

R: We empathize with the reviewer on this point. It is challenging to show 10 different colors in a way that it is easy for the reader to interpret. We did try hard, however, to choose colors that will be (at least when looking at the high-resolution images and zooming in) easy to differentiate. We understand that for certain viewers even this selection may be unsatisfactory, but we propose to maintain these colors for lack of a better alternative.

- "Subsets, such as inflammatory fibroblasts, neutrophils, or inflammatory M1 macrophages ... were absent in HC" A total of 46,700 cells across 18 samples suggests that some of the cell types not identified in HC might be due to low numbers. Therefore some of the results might be biased by the low cell counts.

R: We agree with the reviewer on this important point. As we mention in the Discussion, the reduced number of patients/cells per patient is one of the limitations of this study. Nonetheless, observations such as lack of neutrophils, and other inflammatory subsets in healthy (non-IBD) colon are consistent with the literature on human subjects and with our own unpublished datasets (new Extended Figure 11 for neutrophils). More importantly, the fact that we could detect the rare eosinophil population, even in healthy donors (as seen by scRNA-seq, flow cytometry and IHC), further supports the view that neutrophils may be either absent or very rare in human colonic mucosa.

Reviewer #3 (Remarks to the Author):

In this manuscript by Garrido-Trigo et.al., the authors utilized a combination of scRNA-seq and spatial imaging to identify some interesting phenotypes in macrophages and neutrophils in intestinal tissues that are associated with inflammation. Notable findings include the identification of some novel markers for interesting macrophage populations, as well as the association, based on spatial imaging with inflammatory fibroblasts. Certainly, the results support the notion that macrophage and neutrophil populations are more complexed than previously

appreciated.

However, the main weakness in this study is that the authors do not relate the phenotype of the cellular populations that they observed with the clinical phenotypes of the patients. As the authors note, IBD is marked by perplexing inter-individual variation in clinical phenotype, pathology and response to treatment. The application of single cell and other new technologies has the potential to uncover the basis of inter-patient heterogeneity. But in this study, the authors do not relate the interesting phenotypes they observed with clinical outcomes, or even phenotypes. Hence the overall impact of the study is more limited. Perhaps with some meta-analysis of other datasets, or additional samples, the authors may be able to relate the cellular phenotypes that they have identified with some clinical features of IBD, which may improve the overall manuscript.

R: We thank the reviewer for raising this valuable point. Indeed, our initial objective had been to understand disease heterogeneity by applying single-cell and spatial technologies to patient samples. While this remains our long-term goal, we acknowledge the limitation of our study in addressing the inter-individual variation and agree that much larger-size studies will be required. We estimate that at least a 10-fold increase in the number of individuals studied would be necessary to capture the diversity of features that characterize IBD including disease phenotype, location and severity, treatment at the time of study and time since diagnosis, among other variables. Nonetheless, our study does underscore the usefulness of scRNA-seq and spatial analysis for unraveling the diversity within each cell type and points towards the myeloid (and stromal) compartments as potential key contributors to disease heterogeneity. Despite the limitations of our study, we did observe some tendencies that could be related to disease subtype. For example, we found features such as an abundance of IDA macrophages, which are apparently associated with UC patients (Figure 2b, Milo analysis).

In conclusion, while we cannot yet address the reviewer's important point, we fully agree with its clinical relevance. As these technologies become more affordable and mainstream, we are confident that larger studies will start to provide the molecular basis to understand disease heterogeneity.

REVIEWER COMMENTS

Reviewer #1 (Remarks to the Author):

The authors have done a great job in addressing all of my comments.

Reviewer #2 (Remarks to the Author):

The revised manuscript has substantially improved. The comments are fully addressed or if not reasonably explained.

Reviewer #3 (Remarks to the Author):

While I appreciate the authors honesty in stating that the number of samples analyzed is insufficient to establish the clinical relevance of their observations, to this reviewer, this is also an indication of the limited impact of the study. It may be possible to use the signatures that they observe to perform a meta-analysis of other RNA-seq datasets from IBD patients that have greater numbers with relevant clinical meta-data. There is certainly plenty of RNA-seq or other forms of transcriptional profiling data available, including now substantial samples of scRNA-seq. Ultimately, it's a subjective matter of whether the clinical relevance is an essential component of this manuscript.

Reviewer #1 (Remarks to the Author):

The authors have done a great job in addressing all of my comments.

Reviewer #2 (Remarks to the Author):

The revised manuscript has substantially improved. The comments are fully addressed or if not reasonably explained.

Reviewer #3 (Remarks to the Author):

While I appreciate the authors honesty in stating that the number of samples analyzed is insufficient to establish the clinical relevance of their observations, to this reviewer, this is also an indication of the limited impact of the study. It may be possible to use the signatures that they observe to perform a meta-analysis of other RNA-seq datasets from IBD patients that have greater numbers with relevant clinical meta-data. There is certainly plenty of RNA-seq or other forms of transcriptional profiling data available, including now substantial samples of scRNA-seq. Ultimately, it's a subjective matter of whether the clinical relevance is an essential component of this manuscript.

Response: We thank the reviewer for the effort he/she has devoted to improving the quality of our study. We also agree that having a correlation of specific cell types with important clinical features would be highly desirable. Following the reviewer's important suggestion, we examined the scRNA-seq datasets that are available from IBD patients and which included myeloid cell types. The metadata for each of these sets is shown below (Table 1).

Reference	GSE ID	Gender	Age	Sample Location (ileum/colon)	Disease (UC/CD)	Other
Maddipati, S.C. et al. (2022)	GSE202052	x	x	Terminal ileum	x	Race, time-point
Boland, B.S. et al. (2020)	GSE125527	-	-	Rectum	x	-
Ashton, J.J. et al. (2021)	GSE153866	-	-	Ileum	x	Protocol of digestion
Huang, B. et al. (2019)	GSE121380	-	-	Colon	x	Sample ID, Patient ID, Protocol of cell selection
Devlin, J.T. et al. (2021)	GSE162335	-	-	Pouch/colon	x	-
Mitsialis, V. et al. (2020)	GSE150115	-	-	x	Colon	-
Martin, J. et al. (2019)	GSE134809	-	-	x	Ileum	-

Smillie, C. et al. (2019)	Broad DUOS SCP259	-	-	x	Colon	Sample ID, Patient ID, Protocol of cell selection
Venema, W.T.U. et al. (2019)	EGAS00001002702	*	*	x	Ileum	(*access to this data and metadata is restricted)
Elmentaite, R. et al. (2020)	E-MTAB-8901	-	-	x	-	Sample ID, Patient ID
Kong, L. et al. (2023)	SC Broad Portal/SCP1884	-	-	x	Colon/ileum	-
Elmentaite, R. et al. (2021)	E-MTAB-9543, E-MTAB-9536, E-MTAB-9532, E-MTAB-9533 and E-MTAB-10386	x	x	x	-	Sample ID, organism, developmental stage, cell type

Table 1. Publicly scRNA-seq datasets from IBD intestinal samples including the myeloid compartment.

Unfortunately, beyond sample location (colon or ileum), disease type (CD or UC) and disease activity (inflamed or not inflamed), only a couple of data sets provide age or gender information, while none have variables such as treatment at the time of sampling, disease phenotype (structuring, including perianal fistulae, stenosing or purely inflammatory) in the case of CD, or disease extension for both CD and UC, previous treatments and/or surgeries, intra-intestinal manifestations, time of disease evolution, age at diagnosis, smoking status, or any of the other important variables.

Regarding bulk RNAseq datasets, while it is true that there are numerous public repositories some of them with additional metadata (Table 2).

PAPER	GSE ID	Gender	Age	Sample Location	Disease	Other metadata
Huang, Y. et al. (2017)	GSE81266	x	-	Pouch	x	biopsy time, Patient_ID, prognosis, ethnicity
Lloyd-Price, J. et al. (2019)	GSE111889	x	-	ileum, sigmoid colon, rectum	x	
Haberman, Y. et al. (2019)	GSE101794	x	-	ileum	x	age at diagnosis, paris age
Eshelman, M.A. et al. (2019)	GSE130038	x	-	sigmoid colon	x	smoking history, age at diagnosis, age at surgery, bmi
Haberman, Y., Karns, R., et al. (2019)	GSE109142	x	-	rectal mucosa	x	age at diagnosis, pucai, eosinophil grade >32, week 4 remission, histology severity score, initial treatment, baseline calprotectin, week 4 calprotectin
Mo, A. et al. (2020)	GSE137344	x	-	ileum	x	age at diagnosis, african ancestry

Berger, K. et al. (2021)	GSE158952	x	x	ileum, rectum	x	ancestry
Haberman, Y. et al. (2014)	GSE57945	x	-	ileum	x	paris age, I2 type, histopathology, deep ulcer (yes/no)
Tew, G.W. et al. (2016)	GSE72819	-	-	colonic biopsy	x	Sample_ID, tissue diagnosis, remission week 10, prio anti.tnf, tnf ir (*IBD/non-IBD)
Peters, L.A. et al. (2017)	GSE100833	x	x	rectum, ileum, colon, sigmoid, left colon	*	
Marigorta, U.M. et al. (2017)	GSE93624	x	-	ileum	x	diagnosis, age at diagnosis, Paris, age, ancestry, progression to complication, tissue
Ilott, N.E. et al. (2022)	E-MTAB-9658	x	x	rectum, caecum, ileum	x	Organism, Developmental stage, Sample_ID
Tremblay, E. et al. (2015)	GSE137344	x	-	ileum	x	Age at diagnosis, ancestry

Table 2. Publicly bulk RNA-seq datasets from IBD intestinal samples.

Nonetheless, due to the low-resolution nature of bulk analysis we cannot infer the abundance of populations such as IDA, CXCL5_M1 or ACOD1_M1 macrophages. The genes that separate one macrophage subset from another (NRG1, CXCL5, INHBA or CXCL8, for example) are not exclusively expressed by these subsets of macrophages but are also seen in S2b fibroblasts (for NRG1) or inflammatory fibroblasts (for CXCL5, INHBA or CXCL8) or neutrophils (see Figure below).

Figure. Violin plots showing expression of selected genes in stromal and myeloid cells.

We would like to emphasize that the advantage of single-cell RNA-seq or spatial analysis, compared to bulk RNAseq, is precisely that it allows us to simultaneously measure the combination of genes that define a unique signature by looking at each cell's transcriptome. Without single-cell resolution those signatures do not reveal changes in a specific subset, but rather overall regulation of the gene, which is potentially affected by one or more cell types. Thus, without single-cell resolution we cannot conclude that changes in the expression of those signatures correlate exclusively to a specific lineage (i.e., IDA macrophages) or if these changes stem from several cell types. Furthermore, the bulk RNAseq datasets that are publicly available also contain limited metadata (Table 2), as can be seen with the scRNA-seq ones.

In summary, we want to reassure the reviewer that we have tried to address this point to the best of our ability. Nonetheless, the current datasets cannot provide an answer at this time.